# Role of Incretins in Muscle Functionality, Metabolism, and Body Composition in Breast Cancer: A Metabolic Approach to Understanding This Pathology

**DOI:** 10.3390/biomedicines12020280

**Published:** 2024-01-25

**Authors:** Brenda-Eugenia Martínez-Herrera, Michelle-Guadalupe Muñoz-García, Laura-Liliana José-Ochoa, Luis-Aarón Quiroga-Morales, Luz-María Cervántes-González, Mario-Alberto Mireles-Ramírez, Raúl Delgadillo-Cristerna, Carlos-M. Nuño-Guzmán, Caridad-Aurea Leal-Cortés, Eliseo Portilla-de-Buen, Benjamín Trujillo Hernández, Eduardo Gómez-Sánchez, Martha-Cecilia Velázquez-Flores, Mario Salazar-Páramo, Miguel-Ricardo Ochoa-Plascencia, Daniel Sat-Muñoz, Luz-Ma.-Adriana Balderas-Peña

**Affiliations:** 1Departamento de Nutrición y Dietética, Hospital General de Zona #1, Instituto Mexicano del Seguro Social, OOAD Aguascalientes, Boulevard José María Chavez #1202, Fracc, Lindavista, Aguascalientes 20270, Mexico; bren.mtzh16@gmail.com; 2Carrera de Médico Cirujano y Partero, Coordinación de Servicio Social, Centro Universitario de Ciencias de la Salud (CUCS), Universidad de Guadalajara (UdG), Guadalajara 44340, Mexico; michelle.munoz8626@alumnos.udg.mx (M.-G.M.-G.); laura.jose@alumnos.udg.mx (L.-L.J.-O.); luz.cervantes7983@alumnos.udg.mx (L.-M.C.-G.); 3Becaria de la Dirección General de Calidad y Educación en Salud, Secretaría de Salud Mexico, Comisión Interinstitucional de Formación de Recursos Humanos en Salud, Programa Nacional de Servicio Social en Investigación 2021, Demarcación Territorial Miguel Hidalgo, Ciudad de Mexico 11410, Mexico; 4Unidad Académica de Ciencias de la Salud, Clínica de Rehabilitación y Alto Rendimiento ESPORTIVA, Universidad Autónoma de Guadalajara, Zapopan 45129, Mexico; luisquiroga@hotmail.com; 5División de Investigación en Salud, Unidad Médica de Alta Especialidad (UMAE), Hospital de Especialidades, Centro Médico Nacional de Occidente, Instituto Mexicano del Seguro Social, 1000 Belisario Domínguez, Guadalajara 44340, Mexico; dr_mireles@hotmail.com; 6Unidad Médica de Alta Especialidad (UMAE), Departamento Clínico de Radiologia e Imágen, Hospital de Especialidades (HE), Centro Médico Nacional de Occidente (CMNO), Instituto Mexicano del Seguro Social (IMSS), 1000 Belisario Domínguez, Guadalajara 44340, Mexico; radelcri@outlook.com; 7Unidad Médica de Alta Especialidad (UMAE), Departamento Clínico de Cirugía General, División de Cirugía, Hospital de Especialidades, Centro Médico Nacional de Occidente, Instituto Mexicano del Seguro Social, 1000 Belisario Domínguez, Guadalajara 44340, Mexico; carlosnunoguzman@hotmail.com; 8Servicio de Cirugía General, OPD Hospital Civil de Guadalajara “Fray Antonio Alcalde”, 278 Hospital, Guadalajara 44280, Mexico; 9División de Disciplinas Clínicas, Centro Universitario de Ciencias de la Salud (CUCS), Universidad de Guadalajara (UdG), 950 Sierra Mojada, Building N, 1st Level, Guadalajara 44340, Mexico; eduardo.gsanchez@academicos.udg.mx (E.G.-S.); miguel.oplascencia@academicos.udg.mx (M.-R.O.-P.); 10División de Investigación Quirúrgica, Centro de Investigación Biomédica de Occidente, Instituto Mexicano del Seguro Social, Órgano de Operación Administrativa Desconcentrada, Guadalajara 44340, Mexico; lealc36@yahoo.com.mx (C.-A.L.-C.); eportilla@mail.udg.mx (E.P.-d.-B.); 11Posgrado en Ciencias Médicas, Universidad de Colima, Av. Universidad No. 333, Las Víboras, Colima 28040, Mexico; trujillobenjamin@hotmail.com; 12Cuerpo Académico UDG CA-874 “Ciencias Morfológicas en el Diagnóstico y Tratamiento de la Enfermedad”, 950 Sierra Mojada, Gate 7, Building C, 1st Level, Guadalajara 44340, Mexico; 13Departamento de Morfología, Centro Universitario de Ciencias de la Salud (CUCS), Universidad de Guadalajara (UdG), 950 Sierra Mojada, Gate 7, Building C, 1st Level, Guadalajara 44340, Mexico; c.velazquez@academicos.udg.mx; 14Unidad Médica de Alta Especialidad (UMAE), Departamento Clínico de Anestesiología, División de Cirugía, Hospital de Especialidades, Centro Médico Nacional de Occidente, Instituto Mexicano del Seguro Social, 1000 Belisario Domínguez, Guadalajara 44340, Mexico; 15Academia de Inmunología, Departamento de Fisiología, Centro Universitario de Ciencias de la Salud (CUCS), Universidad de Guadalajara (UdG), 950 Sierra Mojada, Gate 7, Building O, 1st Level, Guadalajara 44340, Mexico; mario.sparamo@academicos.udg.mx; 16OPD Hospital General de Zapopan, Calle Ramón Corona 500, Colonia Centro, Zapopan 45100, Mexico; 17Comité de Cabeza y Cuello, Unidad Médica de Alta Especialidad (UMAE), Hospital de Especialidades, Centro Médico Nacional de Occidente, Instituto Mexicano del Seguro Social, 1000 Belisario Domínguez, Guadalajara 44340, Mexico; 18Unidad Médica de Alta Especialidad (UMAE), Departamento Clínico de Oncología Quirúrgica, División de Oncología Hematología, Hospital de Especialidades, Centro Médico Nacional de Occidente, Instituto Mexicano del Seguro Social, 1000 Belisario Domínguez, Guadalajara 44340, Mexico; 19Unidad de Investigación Biomédica 02 (UIBM 02), Unidad Médica de Alta Especialidad (UMAE), Hospital de Especialidades (HE), Centro Médico Nacional de Occidente (CMNO), Instituto Mexicano del Seguro Social (IMSS), 1000 Belisario Domínguez, Guadalajara 44340, Mexico

**Keywords:** incretins, obesity, breast cancer, muscle quality index, body composition

## Abstract

A poorly studied issue in women with breast cancer is the role of incretins (GIP (glucose-dependent insulinotropic polypeptide) and GLP-1 (glucagon-like peptide-1)) in the quantity and quality of muscle mass in lean and obese individuals. The current report aims to analyze the patterns of association and the role of incretin in muscle functionality and body composition in women with cancer compared with healthy women (mammography BI-RADS I or II) to elucidate whether GIP and GLP-1 can be used to estimate the risk, in conjunction with overweight or obesity, for breast cancer. We designed a case–control study in women with a breast cancer diagnosis confirmed by biopsy in different clinical stages (CS; n = 87) and healthy women with a mastography BI-RADS I or II within the last year (n = 69). The women were grouped according to body mass index (BMI): lean (<25 kg/m^2^BS), overweight (≥25–<30 kg/m^2^BS), and obese (≥30 kg/m^2^BS). We found that GLP-1 and GIP levels over 18 pg/mL were associated with a risk of breast cancer (GIP OR = 36.5 and GLP-1 OR = 4.16, for the entire sample), particularly in obese women (GIP OR = 8.8 and GLP-1 OR = 6.5), and coincidentally with low muscle quality indexes, showed an association between obesity, cancer, incretin defects, and loss of muscle functionality.

## 1. Introduction

Obesity is one of the most important modifiable risk factors for breast cancer, implying various metabolic alterations. It is a common condition in treatment-naïve patients at diagnosis, and it impacts the overall survival in women with this condition across races [1]. Worldwide, breast cancer is one of the most prevalent cancers in women [2]. It affects more women than any other oncologic disease and was one of the leading causes of women’s death between 2000 and 2019 [3]. Gradually, through new target therapies and early diagnosis [4], the epidemiological pattern has been changing, turning breast cancer into a chronic disease with a high rate of functional disabilities, rather than a high-lethality entity, affecting socially active women [5].

In this frame, the confluence of three chronic conditions—obesity, insulin resistance, and breast cancer—profoundly affects the functionality of women with treatment-naïve breast cancer. Their functional status could likely worsen over the course of the disease due to the secondary muscle effects of surgery, chemotherapy, and radiotherapy on the upper limb anatomy and function [6], and it is important to identify those patients who are at risk of major muscle damage alongside treatment.

A poorly studied issue in cancer patients and their nutritional state is the quantity and quality of muscle mass in connection with the presence of sarcopenia and tumor cachexia in cancers with a high metabolic rate; in parallel, some studies based on body composition and its functional characteristics have begun to describe the role of factors such as obesity, muscle mass, and inflammatory serum markers in muscle functionality in patients with obesity-related chronic conditions [7]. To the authors’ knowledge, the relationship among the previously mentioned aspects has not been described in breast cancer patients.

Considering all of the clinical conditions described herein, muscle mass, adiposity, inflammatory serum markers, tissular mediators, and hormones, like incretins, among others, with endocrine, paracrine, and autocrine effects are relevant in functional muscle mass maintenance in patients with breast cancer and obesity in an insulin-resistant environment [8,9].

Incretins are peptide hormones produced by the gastrointestinal tract that are released secondarily to meal nutrients, mainly in response to high loads of carbohydrates, acting as secretagogues to enhance insulin secretion and serum levels. The most important are glucose-dependent insulinotropic polypeptide (GIP) and glucagon-like peptide-1 (GLP-1), secreted from upper K cells and lower L cells in the multicellular glands of the gastrointestinal epithelium after eating [10]. Their role is closely related to the effect of insulin, increasing skeletal muscle microvascular blood flow (MBF) in healthy people [11].

In obese subjects with hyperinsulinism or insulin resistance, the skeletal muscle MBF is altered and even diminished. These alterations are related to changes in the concentrations of C-peptide, glucagon, and the gastrointestinal incretins GIP and GLP-1 in response to the ingestion of a high load of oral carbohydrates, as expressed by the MBF changes in all skeletal muscles [11].

On the basis of these physiological and pathophysiological changes in the skeletal muscles, handgrip strength (HGS) is considered a functional indicator of nutritional status and muscle functionality. Some authors have proposed its use as a universally standardized tool to evaluate muscle health in vulnerable population groups with sarcopenia risk, given that it can be a potent predictive marker for disability, morbidity, and mortality [12,13,14,15].

In breast cancer, the overweight–obesity phenomenon has been described as a risk factor and as a parallel condition during diagnosis and treatment. The current evidence suggests that HGS can be reduced in obese people and constitutes a mortality predictor. In this scenario, the muscle quality index, obtained from the relationship between HGS and BMI, indicates health status and physical functioning, expressed as kg/BMI. Some authors suggest that a low muscle quality index (MQI), like HGS, is an independent risk factor for mortality and metabolic complications, including sarcopenia in some malignant neoplasms [16,17,18,19,20,21,22].

A chronic state of low-grade inflammation related to obesity involves all metabolic phenomena in insulin resistance. The inflammatory phenomena affect immune cell infiltration and inflammatory cytokine production in adipose, liver, muscle mass, and pancreatic tissues, with the dysfunction acting in a synergistic manner with the incretin defects and their profound effects on skeletal muscle tissue [23].

The current report aimed to analyze the patterns of association and the role of incretins, muscle functionality, and body composition in women with cancer compared with healthy women (mammography BI-RADS I or II), trying to elucidate whether GIP and GLP-1 can estimate the risk, in conjunction with overweight or obesity, for breast cancer.

## 2. Materials and Methods

Our organization’s Institutional Review Board (Research and Ethics Local Committee 1301) approved the study. We performed a case–control study with the following selection criteria: (A) Inclusion criteria—women with a breast cancer diagnosis confirmed by biopsy in different clinical stages (CS; n = 87) and healthy women who received a mastography BI-RADS I or II within the last year (n = 69). (B) We did not include women with previous cancer, autoimmune diseases, chronic lung disease, cardiovascular diseases, renal failure, or contraindications to the analysis of body composition using electrical bio-impedance. All participants signed an informed consent form to participate.

Women were grouped by body mass index (BMI): lean (<25 kg/m^2^BS), overweight (≤25–<30 kg/m^2^BS), and obese (≤30 kg/m^2^BS).

We measured serum levels of C-peptide, incretins, glucagon, PAI-1 (plasminogen activator inhibitor-1), adipocytokines, glucose, and insulin after calculating the HOMA-IR (HOMA index for insulin resistance) to evaluate the metabolic environment in the studied patients.

On the basis of our previously published data obtained from healthy women over 40 years old, we took the reference values for HOMA-IR (0.78), insulin (2.6 mU/mL), leptin (27.5 ng/mL), adiponectin (17.68 mg/mL), resistin (0.59 ng/mL), visfatin (1.18 ng/mL), and adipsin (0.91 mg/mL) [24].

### 2.1. Body Composition Analysis by BIA, Anthropometric Measurements, Muscular Strength, Body Indexes, and Muscle Quality Calculations Adjusted According to Body Surface

The women included in the study were measured for the body composition analysis conducted using BIA (bio-impedance analysis) after eight hours of fasting, barefoot with clean and dry feet, and without metal jewelry or electrical devices in their bodies. We measured height using an SECA stadiometer (SECA^®^, Hamburg-Landsec, Germany) and weight and body composition (lean mass, fat mass, and skeletal muscle mass (kg)) with a BF-601F BIA device (TANITA^®^, Tokyo, Japan). The body mass index (BMI), fat mass index (FMI), and skeletal muscle mass index (SMMI) were calculated by considering the total weight in kilograms from the total body weight, fat kilograms, and sum of lean mass for the upper and lower limbs divided by the height squared. We determined muscular strength using a Jamar Plus+ digital hand dynamometer (Patterson Medical Supply, Cedarburg, WI, USA) according to the recommendations of the American Association of Hand Therapists [25]. The patients held the device and compressed it with maximum force to obtain a maximum contraction. The test was repeated three times for each hand, with one-minute rest intervals between measurements. The highest result among the tests was recorded. The obtained handgrip strength was divided by the BMI, SMMI, and FMI to obtain the muscle quality index concerning the mentioned anthropometric indexes, which are expressed in kg/BMI, kg/SMMI, and kg/FMI, respectively [7]. To establish normal values and cutoffs for the different variants of the muscle quality index, we used the mean and standard deviation of each index as observed in healthy–lean women.

### 2.2. Blood Sample and Metabolic Markers and Adipocytokines Levels

A blood sample was extracted from patients after 9 to 12 h of fasting. The fasting instructions did not state that the study subjects could not drink water. Peripheral venous blood was extracted from basilic or cephalic arm veins. The glucose level was determined in serum samples using the VITROS^®^ 350/5600 chemistry system from Ortho-Clinical Diagnostics (Johnson & Johnson, New Brunswick, NJ, USA). Five milliliters of venous blood was collected to obtain the serum by centrifugation. We processed the serum samples according to standard guidelines published by the BIO-RAD diabetes panels I and II for human serum (multitest panel Bio-Plex Pro No. 171A7001M and Bio-Plex Pro 171A7002M, BIO-RAD^®^, Hercules, CA, USA) to determine the C-peptide, ghrelin, GIP, GLP-1, glucagon, insulin, leptin, PAI-1, resistin, and visfatin serum levels. The Bio-Plex Pro™ human diabetes 10-plex and 2-plex immunoassays are 1 × 96-well kits that include coupled magnetic beads, detection antibodies, standards, assay buffer, wash buffer, sample diluent, standard diluent, detection antibody diluent, streptavidin-PE, a filter plate, flat-bottom plate, and sealing tape. The assays are based on 6.5 µm magnetic beads, and are compatible with magnetic wash stations. Magnetic washing simplifies plate processing, and it provides increased throughput and results in decreased variability and increased reproducibility. The HOMA-IR was calculated with the reported serum glucose and insulin concentrations [26]. The diagnosis of insulin resistance applies to women with HOMA-IR > 2.6 (values for Mexican women [27]).

### 2.3. Statistical Analysis

We performed the statistical analysis in three phases:

(1) Descriptive: The biochemical values with parametric behavior are described using the mean and standard deviation. For the descriptive analysis, a description of the biochemical markers with nonparametric behavior using the median and interquartile range is provided. Qualitative characteristics, such as overweight, obesity, diagnosis group, and clinical stage, are analyzed by calculating the proportions and percentages.

(2) Inferential: In this phase, we compared the biochemical values according to diagnostic group and nutritional diagnosis (i.e., lean, overweight, and obese women with and without cancer) using ANOVA, and for each in-group the adjusted Bonferroni test to compare the variables with a parametric data distribution among the groups. We performed the Kruskal–Wallis and adjusted Bonferroni tests for the in-group analysis of the variables with a nonparametric distribution. We obtained values for the Hedges’ G and Cohen or eta and epsilon squared for the numerical variables with a parametrical distribution, and we identified statistical differences using the means and standard deviations between 0.23 and 0.5 to estimate the sample size effect. We calculated a chi-square or Fisher’s exact test if the expected values were less than five for differences between a specific percentage in one clinical condition.

(3) Association analysis: Relationships among the variables were identified using Pearson’s correlation (quantitative variables) or Spearman’s rho (ordinal or relation between ordinal and quantitative variables); those with a significance < 0.2 were selected and used in a logistic regression model to obtain the OR with 95% confidence intervals (CI95%). Finally, we constructed the model considering two health state conditions—(1) healthy women and (2) women with breast cancer—and three nutritional diagnoses: (A) lean women, (B) overweight women, and (C) obese women.

(4) Multivariate analysis: As part of this analysis, we established normal values and cutoffs for the following variables: C-peptide, GIP, GLP-1, and PAI-1. The above-mentioned values are provided in Section 3 (Results). On this basis, we dichotomized the variables to generate a logistic regression analysis in the first step for the sample’s total, considering the control subjects and cancer patients to estimate the OR for developing cancer in our sample control group of women. As part of the logistic regression, we correlated the variables to select pairs of variables with a correlation *p*-value below 0.2 for use in the logistic regression model. Using this method, we introduced, step by step and in descending order, the variables with the highest correlation values. When we noticed a change in the significance of the model, we stopped the analysis and changed the order of the introduction of the variables to identify if a loss of significance revealed confounding variables. In the final phase, we evaluated the OR grouping of the women according to their nutritional status in normal, overweight, and obese groups.

## 3. Results

We studied 156 patients from the breast cancer screening program in care at a Mexican tertiary multidisciplinary facility; 69 were healthy women with a mammography classified as BI-RADS I or II, and 87 had breast cancer in diverse clinical stages.

The women were studied according to their health status and nutritional diagnosis:A.Healthy women (n = 69):
a.Normal weight (n = 22, 31.9%);b.Overweight (n = 32, 46.4%);c.Obese (n = 15, 21.7%).
B.Women with breast cancer (n = 87):a.Normal weight (n = 26, 29.9%);b.Overweight (n = 28, 32.2%);c.Obese (n = 33, 37.9%).

The mean age of the healthy women was 50.7 (±9.3) years old, and for the breast cancer group it was 53.47 (±11.15) years old, with no statistical difference between the groups. Regarding the progression of the disease, we did not find significant differences based on the clinical stage. The overweight/obesity rates were high in both groups (68% in healthy women and 70.1% in women with breast cancer; *p* = NS).

In our previously published article regarding normal adipocytokine values in healthy Mexican women over the age of 40 [24], we determined the results according to the following analytes and subrogated variables: leptin, adiponectin, resistin, visfatin, adipsin, and HOMA-IR (insulin = 2.6 mU/mL, leptin = 27.5 ng/mL, adiponectin = 17.68 mg/mL, resistin = 0.59 ng/mL, visfatin = 1.18 ng/mL, and adipsin = 0.91 mg/mL, HOMA-IR = 0.78) [21]. The current report determined the normal values based on the median and interquartile intervals for the key metabolic mediators and anthropometric characteristics (Table 1).

### 3.1. Anthropometric Parameters in Healthy Women and Women with Breast Cancer Groups by Nutritional State

#### Age and Anthropometric Indexes

In both the healthy women and women with breast cancer groups, the mean age was approximately 50 years, and the nutritional state was not influenced by age. Concerning the anthropometric indexes, we found statistical differences between healthy women and women with breast cancer in the appendicular skeletal muscle index (ASMMI), fat mass index (FMI), and fat mass percentage (%) when we classified them in an intragroup analysis according to their nutritional state by BMI (see Table 2). We used the SMMI results to look for sarcopenia; no sarcopenic women were identified in our study groups (i.e., control or cancer groups).

### 3.2. Handgrip Strength and Muscle Quality Adjusted for BMI, ASMMI, and FMI

Our analysis showed no significant difference in handgrip strength regarding nutritional status (i.e., normal weight, overweight, and obese) (Table 2) when the women in the healthy group and breast cancer group were studied separately. Women who were overweight showed differences in the HGS and MQI adjusted for BMI, appendicular skeletal muscle mass index (ASMMI), and fat mass index (FMI) (see Table 2).

On the basis of the cutoffs for the normal values, as described in Table 1, for fat mass percentage, fat mass index (FMI), max HGS, and the MQI adjusted for BMI, SMMI, and FMI, we compared the percentage of healthy women and women with breast cancer that had normal or abnormal values for each anthropometric parameter, and we found differences in the percentage of women with a normal max HGS, MQI adjusted for BMI and SMMI, and a normal FMI in overweight women in the healthy women group and women with breast cancer group, with a lower max HGS and lower MQI adjusting for the three above-mentioned anthropometric indexes (see Table 3).

### 3.3. Comparison of Adipocytokines and Metabolic Biomarkers between the Health and Nutritional States

Lower levels of ghrelin were observed for noncomorbidities in women with overweight and obesity compared with healthy normal-weight women (71.7% versus 65% and 48.6%, respectively). However, there were no significant differences in glucagon, insulin, and HOMA-IR between the different nutritional states. The levels of C-peptide, GIP, adiponectin, resistin, visfatin, PAI-1, and adipsin were also not significantly different among the groups. However, the levels of leptin showed a differentiated pattern, with the lowest values in the healthy normal-weight women (1.6 (0.6–3.2); *p* = 0.007, compared to other nutritional states in healthy women) and the highest values in the group with breast cancer and obesity (12.23 (15.31) ng/mL; *p* = 0.014, compared to other nutritional states in women with breast cancer, see Table 4). These findings are consistent with our previous reports.

After analyzing the reference value cutoffs, we discovered that the breast cancer patients who were normal weight and those who were overweight exhibited altered levels of specific biomarkers. However, in obese women, this effect appears to be less remarkable. Despite this, when we calculated the raw odds ratio, we found that C-peptide, GIP, GLP-1, and PAI-1 are indicators for breast cancer risk (Table 5).

### 3.4. Comparison of Anthropometric Parameters by Health State

During the second phase of our analysis, we observed a significant difference in the handgrip strength variable between women in the control and breast cancer groups. In women with normal weight, handgrip strength was markedly lower in breast cancer patients (21.57 (±4.12)) compared with the control group (25.5 (±5.4)), with a *p*-value of 0.007. We also noticed a similar trend in the muscle quality index adjusted for BMI, although the difference was not statistically significant (*p* = 0.056).

In the subgroup of overweight women, there were significant differences between BMI, the maximum handgrip strength, and the muscle quality index adjusted for BMI, ASMMI, and FMI. The women with breast cancer group had lower values for muscle functional indicators. In obese women with and without cancer, we did not observe differences for age or any anthropometrical parameter (see Table 6).

However, in the group of obese women, no significant differences were found between those with breast cancer and those without (*p*-values > 0.05).

The study found a significant difference in the HGS and MQI adjusted for BMI in normal-weight women with and without cancer when compared to women who were overweight.

### 3.5. Comparison of Adipocytokines Levels by Health State

Concerning the adipocytokine values in normal-weight women, the data show profound differences between the healthy women and women with breast cancer groups, with lower values observed in the control group for C-peptide, GIP, GLP-1, ghrelin, glucagon insulin, HOMA-IR, leptin, and adiponectin (*p* < 0.05) and the highest values for PAI-1. We did not observe differences in the levels of resistin, visfatin, and adipsin (*p* = NS).

In obese patients, we observed statistical differences in the levels of C-peptide, GIP, ghrelin, insulin, HOMA-IR, adiponectin, resistin, and PAI-1. However, the differences observed for the women in the control group who were normal weight and overweight with respect to the breast cancer groups were not evident in obese women (see Table 7).

Concerning the relationships among the variables and how they affect their role as risk factors, we found that elevated levels of GIP and GLP-1 are significant risk estimators in both the general population and obese women (see Table 8).

## 4. Discussion

Both sarcopenia and tumor cachexia are common clinical characteristics in most oncologic diseases. However, despite previous descriptions by Chen L.K. et al., Morton M. et al., and Seke E.P.F. et al., we did not observe any such phenomena in our study despite the enhancement of pro-inflammatory adipocytokines in the selected patients [28,29,30]. Upon comparing the functional parameters related to muscle function, we found significant damage in breast cancer patients compared with women in the control group. Along with the detrimental effects on muscle function, we observed a rise in gastrointestinal incretins (GIP and GLP-1), C-peptide, and glucagon levels, along with low PAI-1 levels, in breast cancer patients, independently. Our current data show converse results with respect to previous reports by Jujic A. et al. and are partially coincidental with the theory presented by Samuel SM, et al. [31,32].

The altered muscle function in patients with cancer and chronic disease often leads to falls, disability, and functional impairment, increasing the risk of a poor outcome and death [33]. We calculated the muscle quality index (MQI) by measuring muscle strength using a handgrip dynamometer and normalizing it with either the body mass index or appendicular skeletal muscle mass index (ASMMI) [7].

Authors such as Caamaño-Navarrete et al., have emphasized the importance of the MQI, even when based only on BMI (MQI/BMI), as an independent factor that increases disability and mortality rates [13,17]. Some studies indicate that several diseases related to metabolic syndrome (MetS) are associated with low MQI/BMI and increased mortality due to insulin resistance, which has an inverse correlation with insulin sensitivity [34]. Our study found that individuals with obesity exhibited a worse MQI, which might be linked to insulin resistance and high values in the MetS parameters [35,36], with statistically significant correlation coefficients in both cases, a condition observed in our current report on obese women with and without cancer [34].

In their study, Roberti et al. found that despite a loss of muscle mass, the muscle quality index (QMI) is a reliable predictor of prognosis. Our report shows that a high percentage of breast cancer patients have a deteriorated QMI when adjusted for BMI, ASMMI, and FMI. This condition is even more evident in overweight and obese women, and although we did not find sarcopenia or cancer cachexia, the results indicate a need to monitor muscle quality in breast cancer patients to improve outcomes [37]. The authors concluded that intermuscular adipose tissue may be responsible for shorter progression-free survival, leading to changes in muscle quality without affecting the muscle quantity and worsening the patient’s prognosis [13].

Although the definition of sarcopenia varies depending on each author [38], it generally refers to the loss of skeletal muscle mass and function that occurs secondarily to factors such as chronic disease and older age. However, in patients with obesity or metabolic syndrome, for example, biochemical and molecular changes can affect the functionality of muscle fibers even if the person maintains their muscle mass, as Caamaño et al. described [7].

The muscle quality index (MQI) has become popular as an effective tool for assessing adult muscle functionality in recent years. When combined with gait speed and handgrip strength (HGS), the MQI can accurately evaluate muscle functionality in older patients with chronic diseases. This is particularly helpful when measuring the impact of a training program on an individual’s functional state [21].

After analyzing our patient group, we found a specific occurrence of normal values in the skeletal muscle mass and appendicular skeletal muscle mass indexes. However, we noticed a change in the MQI among women with cancer and obese women with and without cancer, which was linked to high levels of C-peptide, glucagon, and gastrointestinal incretins (GIP and GLP-1).

Physiological studies on healthy individuals have demonstrated that intravenous insulin infusion increases blood flow in skeletal muscle microvessels. However, this effect is not observed in individuals with insulin resistance. When glucose is administered orally, the phenomenon is altered in obese individuals or those with insulin resistance, even if they have an adequate BMI.

Hormones produced by the gut may have an impact on this effect. To support this claim, Roberts-Thompson et al. conducted a 120 min oral glucose tolerance test (OGTT) using 75 g of glucose [11]. The results showed a different metabolic pattern compared with an intravenous glucose tolerance test (IVGTT), which bypasses the gastrointestinal barrier. Although both tests achieved similar blood glucose concentrations, the OGTT test suggests that hormones derived from the gut may play a role in this effect [7,11], and we observed higher levels of GIP and GLP-1 in obese cancer women with lower muscle function according to the MQI adjusted for BMI and by SMMI.

Likewise, Roberts-Thompson et al. conducted a study in which they measured the plasma levels of various substances such as insulin, C-peptide, glucagon, nonesterified fatty acids, and gut-derived hormones and incretins (gastric inhibitory polypeptide (GIP) and glucagon-like peptide-1 (GLP-1)) before and after OGTT/IVGTT. They found that during the IVGTT the microvascular blood flow (MBF) in skeletal muscle was preserved, whereas during the OGTT, the MBF was impaired [11]. 

The changes in the blood flow of skeletal muscles occurred during an oral glucose tolerance test (OGTT) despite the increased insulin levels, described by Roberts-Thompson, which could be present in obese women with breast cancer. However, more extensive clinical research is required to confirm the proposed mechanism. These changes were negatively correlated with the differences in the GIP and GLP-1 concentrations, while the GLP-1:GIP ratio was positively associated with changes in skeletal muscle blood flow. The results suggest that insulin, GLP-1, and GIP have opposite effects on the microvasculature of skeletal muscles [11].

Furthermore, the study highlights the potential role of GLP-1 and GIP in modulating skeletal muscle blood flow [11], and their data are coincidental with our results, as we observed a deteriorated MQI in association with high levels of gastrointestinal incretins.

Nevertheless, the microvascular responses during postprandial hyperglycemia do not only depend on glucose levels. It seems that the difference between the oral glucose tolerance test (OGTT) and the intravenous glucose tolerance test (IVGTT) is due to increased incretins in the gastrointestinal epithelium. This effect triggers a mechanism through which insulin or mixed-nutrient meal loads stimulate skeletal muscle MBF to contribute to the delivery of nutrients. However, this vascular effect is no longer present in insulin-resistant patients [11].

Consuming food and drinks high in glucose damages skeletal muscle MBF in healthy individuals and those with obesity or insulin resistance, as observed in our women’s groups. However, when glucose is administered intravenously, it bypasses the gastrointestinal tract and inhibits the actions and production of incretins. The authors highlight the importance of the impact of high glucose levels on the effects of incretins, which may lead to skeletal muscle MBF damage and negatively affect vascular muscle health. This condition may be present in patients with breast cancer and high rates of overweight and obesity [11].

Maintaining a stable glucose metabolism is essential to sustain the anabolic phase of metabolism. The hormone insulin plays a crucial role in regulating glucose production from the liver and kidneys and balancing glucose disposal in peripheral tissues such as skeletal muscle. Adipose tissue also significantly participates in this process by providing nonesterified fatty acids (NEFAs) as an alternative metabolic substrate for skeletal muscle cells and hepatocytes when glucose levels are low [39], metabolic conditions that could explain the phenomena observed in our group of women with breast cancer.

During nocturnal fasting, the body gradually develops insulin resistance due to the secretion of growth hormone and cortisol. This resistance helps in maintaining glucose levels through two mechanisms: firstly, by switching glucose to NEFA oxidation in muscle cells, and, secondly, by balancing endogenous glucose production in the liver and kidneys [39].

In DM2-related metabolic disorders, there are differences in the metabolic mediators such as gastrointestinal incretins, C-peptide, glucagon, and PAI-1. These differences lead to a decrease in peripheral glucose uptake, particularly in muscle cells. Moreover, there is an increase in endogenous glucose levels, insulin resistance, enhanced lipolysis, and production of free fatty acids, as well as the accumulation of intermediate lipids that favor the increase in glucose output. Because of all these changes, there is a reduced peripheral use of glucose and altered beta-cell functioning [40].

Glucotoxicity, lipotoxicity, and pancreatic beta-cell damage due to the long-term elevation of glucose and lipid levels in the body are associated with both insulin resistance and inflammation, which are significant factors in certain types of cancer development, resulting in insulin resistance at the tissue level, which adds to the abnormalities in pancreatic beta-cell functioning. Compensatory insulin secretion worsens over time, contributing to further pancreatic function deterioration, cancer, and related disease progression [40].

The previously mentioned phenomena are associated with damaged glucagon release by the pancreatic alpha cells. This is more evident during postprandial phases with simultaneous alterations in the insulin secretion pattern and excessive glucagon secretion due to an incretin defect. Specifically, there is an inadequate response to GIP and GLP-1 after meals [40], exhibiting a biochemical pattern similar to that observed in obese women with breast cancer.

For women with breast cancer who have insulin resistance and dysmetabolic patterns, these conditions could increase the risk of obesity or insulin resistance [40].

Within muscle tissue, insulin receptors are part of the tyrosine kinase family, which can phosphorylate specific intracellular proteins—IRS-1, IRS-2, IRS-3, and IRS-4—which are four identified proteins [41] that act as substrates and interact with the insulin receptor tyrosine kinase (primarily IRS-1). Genetic or posttranscriptional changes that occur can impair the insulin’s ability to stimulate glycogen or DNA synthesis, thereby highlighting the crucial role of IRS-1 in insulin signal transduction [42].

In muscle tissue, phosphorylated tyrosine residues on IRS-1 activate phosphatidylinositol 3-kinase to stimulate glucose transport and glycogen synthase [43].

The insulin, in turn, increases protein synthesis and inhibits their degradation through PI-3 kinase, activating mTOR. This molecule controls translation mechanisms by phosphorylation and activation of p70 ribosomal S6 kinase (p70rsk), phosphorylating initiation factors. Insulin increases the transcription factor SREBP-1c, mediated through the PI3–kinase pathway, stimulating the liver to synthesize triglycerides [44].

When insulin is present, proteins containing SH2 domains (namely, the adapter proteins Grb2 and Shc) interact with IRS-1, leading to their phosphorylation. This interaction serves as a link between IRS-1/IRS-2 and the mitogen-activated protein (MAP) signaling pathway, which, in turn, generates various transcription factors. Following this, Ras is activated, and a cascade of events leads to the stepwise activation of Raf, MEK, and ERK. Once ERK is activated, it moves into the cell nucleus and catalyzes the phosphorylation of other transcription factors. This mechanism enables insulin to promote cell growth, proliferation, and cell differentiation [40]. In other words, blocking the MAP kinase pathway could potentially stop cell growth while leaving the metabolic actions of the hormone unaffected, which would be particularly relevant in cases of breast cancer or colorectal cancer, where insulin promotes cell growth and proliferation in malignant clones. These conditions have been observed previously in our reports concerning insulin and adipocytokines in obese women with breast cancer [24].

During anabolism, insulin triggers glycogen synthesis by activating glycogen synthase while inhibiting glycogen phosphorylase. The PI3-k pathway mediates insulin’s effect by activating phosphatases, such as protein phosphatase 1 (PP1), which is the primary regulator of glycogen. In skeletal muscle, PP1 is associated with a specific glycogen-binding regulatory subunit, leading to the dephosphorylation of glycogen synthase and the activation of the glycogen synthase pathway [40].

Incretins, such as GIP and GLP-1, are essential in regulating insulin secretion in response to nutrient intake. However, in situations in which obesity and insulin resistance are present, their effect may be missing, reduced, or impaired because of their interaction with muscles and splanchnic tissues [23].

Obesity and insulin resistance often lead to high blood sugar levels. Eight metabolic issues contribute to this, decreased peripheral glucose uptake, increased hepatic glucose production, increased lipolysis, reduced peripheral use of glucose in muscle cells, compensatory insulin secretion, unappropriated glucagon release, inadequate secretion of gastrointestinal incretin hormones, and enhanced renal tubular glucose reabsorption, such as we observed in obese women with breast cancer [24].

Insulin resistance can cause a chronic low-grade inflammatory state that affects various organs, including adipose tissue, liver, muscle mass, and pancreatic tissues. This results in immune cell infiltration and the production of inflammatory cytokines, leading to dysfunction in these areas. All eight phenomena mentioned are part of the body’s nonspecific immune response to insulin resistance [40].

## 5. Conclusions

Obesity and insulin resistance are metabolic conditions commonly related to breast cancer development and progression. Microenvironmental conditions trigger a phenomenon known as incretin defects, which causes muscle dysfunction secondary to the alteration of skeletal muscle MBF, according to the results of previous physiological studies. In our patients, we observed muscle and incretin alterations in obese women with breast cancer, coincidentally with data found in those functional studies; additionally, we estimated the risk of developing cancer and its relationship with incretin alterations, and we found that levels over 18 pg/mL for GLP-1 and GIP are associated with an OR of the risk for breast cancer (GIP OR = 36.5 and GLP-1 OR = 4.16, for the entire sample), particularly in obese women (GIP OR = 8.8 and GLP-1: OR = 6.5), in a manner coincidental with low muscle quality indexes, showing an association between obesity, cancer, incretin defects, and loss of muscle functionality.

Incretins’ role in cancer development has been mentioned as part of the spectrum of metabolic implications related to obesity, such as modifiable cancer development risk factors. However, there exists in breast cancer research the issue of limited information concerning incretins’ role in the metabolic pathophysiology of the disease, so we consider the results presented as part of this paper to be valuable for the establishment of an integral way to treat breast cancer and the co-morbid metabolic alterations related to it. 

The findings of our current report must be considered as part of the spectrum of deleterious metabolic effects tied to obesity. At this stage, it is a current priority to enhance multidisciplinary strategies for the treatment of breast cancer survivors via a multimodal method that includes close nutritional follow-up and is associated with force and resistance training programs oriented toward preserving and enhancing skeletal muscle mass and its functionality, as part of standardized good clinical practices in treating breast cancer survivors. In parallel, the biochemical follow-up must include metabolic indicators such as glucose, glycosylated hemoglobin, HOMA-IR, insulin, C-reactive protein, serum lipids, and incretins.

## Figures and Tables

**Table 1 biomedicines-12-00280-t001:** Biochemical (values for C-peptide, GIP, GLP-1, ghrelin, glucagon, and PAI-1 levels were determined in serum samples) and anthropometric reference values from 69 metabolically healthy Mexican women over 40 years old (reference values).

	Normal Values *	Cutoff **
**Biochemical reference values**		
C-peptide (pg/mL)	178 (149–216)	>216
GIP (pg/mL)	15 (11–18)	>18
GLP-1 (pg/mL)	15 (11–18)	>18
Ghrelin (pg/mL)	72 (66–106)	>106
Glucagon (pg/mL)	186 (155–192)	>192
PAI-1 (ng/mL)	266 (94–118)	<94
**Anthropometric reference values**		
Fat Mass Percentage (percentage%)	32 (28–36)	>36
Fat Mass Index (kg/m^2^BS)	8 (6–8.5)	>8.5
Maximum Grip Strength (kg)	26 (23–27)	<26
Muscle Quality Index—BMI (kg/BMI)	1.08 (0.9–1.3)	<0.9
Muscle Quality Index—SMMI (kg/SMMI)	4 (3.3–4.3)	<3.3
Muscle Quality Index—FMI (kg/FMI)	3.4 (3–4.2)	<3

* Median and interquartile intervals. ** Cutoff values.

**Table 2 biomedicines-12-00280-t002:** Comparison of the anthropometric parameters between healthy women and women with breast cancer grouped by nutritional diagnosis.

	Healthy Women (Control Group, n = 69)				Women with Breast Cancer (Case Group, n = 87)			
Age and Anthropometrical Indicators	Normal Weight n = 22 Mean (SD)	Overweight n = 32 Mean (SD)	Obesity n = 15 Mean (SD)	*p*-Value *	Normal Weight n = 26 Mean (SD)	Overweight n = 28 Mean (SD)	Obesity n = 33 Mean (SD)	*p*-Value *
Age (years)	48.91 (12.36)	51.5 (6.64)	51.6 (9.24)	0.558	51.27 (11.82)	54.14 (9.84)	54.64 (11.71)	0.483
Body Mass Index (kg/m^2^)	23.2 (1.94) *	26.88 (1.09) *	34.59 (4.17) *	**0.000**	22.34 (1.56) *	27.98 (1.22) *	34.35 (4.28) *	**0.000**
Appendicular Skeletal Muscle Index (kg/m^2^)	6.46 (0.45) *	6.92 (0.42)	7.89 (0.94) *	**0.000**	6.22 (0.55) *	7.04 (0.75) **	7.97 (0.94) *	**0.000**
Fat Mass Index (kg/m^2^)	7.18 (2.25) *	9.93 (1.42) *	15.49 (3.25) *	**0.000**	6.91 (1.25) *	10.55 (1.17) *	15.20 (2.95) *	**0.000**
Fat Mass Percentage (%)	30.63 (8.43) *	36.91 (4.87) *	44.43 (4.80) *	**0.000**	30.78 (4.21) *	37.64 (3.32) *	43.98 (3.71) *	**0.000**
Max Handgrip Strength (kg)	25.48 (5.47)	26.5 (4.81)	24.04 (4.04)	0.277	21.56 (4.12)	22.52 (6.49)	22.01 (5.54)	0.817
Muscle Quality Index According to Max Handgrip Strength	1.1 (0.25) *	0.98 (0.16)	0.7 (0.13) *	**0.000**	0.97 (0.21)	0.8 (0.23) *	0.64 (0.18) *	**0.000**
Muscle Quality Index According to Appendicular Skeletal Muscle Index	3.95 (0.85) *	3.82 (0.64) *	3.07 (0.59) *	**0.001**	3.51 (0.83) *	3.21 (0.96)	2.78 (0.73) *	**0.005**
Muscle Quality Index According to Fat Mass Index	2.66 (0.81) *	2.87 (1.09) *	1.85 (0.93) *	**0.005**	3.35 (3.68)	2.16 (0.95)	2.34 (0.98)	0.100

* Significant with a *p*-value < 0.05. The ANOVA and Bonferroni tests were used to identify intragroup differences.

**Table 3 biomedicines-12-00280-t003:** Comparison of the anthropometric parameters between healthy women and women with breast cancer, with normal and abnormal values.

**Normal-Weight Women (n = 48)**			
**Age and Anthropometrical Indicators**	**Control Group** **n = 22** **n** **(%)**	**Breast Cancer Group** **n = 26** **n (%)**	** *p* ** **-Value ***
Fat Mass Percentage			
<36% (normal fat mass percentage)	16 (76.2%)	25 (96.2%)	
≥36%	5 (23.8%)	1 (3.8%)	0.076
Fat Mass Index			
<8.5 kg/m^2^BS (normal fat mass index)	14 (66.7%)	25 (96.2%)	
≥8.5 kg/m^2^BS	7 (33.3%)	1 (3.8%)	**0.015**
Handgrip Strength			
≥26 kg (normal HGS)	9 (42.9%)	3 (11.5%)	
<26 kg	12 (57.1%)	23 (88.5%)	**0.020**
Muscle Quality Index (MQI) Adjusted for BMI			
≥0.9 kg/kg/m^2^BS (normal MQI/BMI)	18 (85.7%)	18 (69.2%)	
<0.9 kg/kg/m^2^BS	3 (14.3%)	8 (72.7%)	0.300
Muscle Quality Index (MQI) Adjusted for SMMI			
≥3.3 kg/kg/m^2^BS (normal MQI/SMMI)	16 (76.2%)	15 (57.7%)	
<3.3 kg/kg/m^2^BS	5 (23.8%)	11 (42.3%)	0.226
Muscle Quality Index (MQI) Adjusted for FMI			
≥3 kg/kg/m^2^BS (normal MQI/FMI)	16 (76.2%)	16 (61.5%)	
<3 kg/kg/m^2^BS	5 (23.8%)	10 (38.5%)	0.355
**Overweight Women (n = 60)**			
**Age and Anthropometrical Indicators**	**Control Group** **n = 32** **n (%)**	**Breast Cancer Group** **n = 28 ** **n (%)**	** *p* ** **-Value ***
Fat Mass Percentage			
<36%	12 (36.4%)	7 (25%)	
≥36%	21 (63.6%)	21 (75%)	0.412
Fat Mass Index			
<8.5 kg/m^2^BS	4 (12.1%)	1 (3.6%)	
≥8.5 kg/m^2^BS	29 (87.9%)	27 (96.4%)	0.363
Handgrip Strength			
≥26 kg	21 (63.6%)	7 (25%)	
<26 kg	12 (36.4%)	21 (75%)	**0.004**
Muscle Quality Index (MQI) Adjusted for BMI			
≥0.9 kg/kg/m^2^BS	26 (78.8%)	8 (28.6%)	
<0.9 kg/kg/m^2^BS	7 (21.2%)	20 (71.4%)	**0.000**
Muscle Quality Index (MQI) Adjusted for SMMI			
≥3.3 kg/kg/m^2^BS	26 (78.8%)	9 (32.1%)	
<3.3 kg/kg/m^2^BS	7 (21.2%)	19 (67.9%)	**0.000**
Muscle Quality Index (MQI) Adjusted for FMI			
≥3 kg/kg/m^2^BS	10 (30.3%)	1 (3.6%)	
<3 kg/kg/m^2^BS	23 (69.7%)	27 (96.4%)	**0.008**
**Obese Women (n = 48)**			
**Age and Anthropometrical Indicators**	**Control Women** **n = 15 ** **n (%)**	**Women with Breast Cancer** **n = 33** **n (%)**	** *p* ** **-Value ***
Fat Mass Percentage			
<36%	-	-	
≥36%	15 (100%)	33 (100%)	-
Fat Mass Index			
<8.5 kg/m^2^BS	-	-	
≥8.5 kg/m^2^BS	15 (100%)	33 (100%)	-
Handgrip Strength			
≥26 kg	4 (26.7%)	9.8 (27.3%)	
<26 kg	11 (73.3%)	24 (72.7%)	1.000
Muscle Quality Index (MQI) Adjusted for BMI			
≥0.9 kg/kg/m^2^BS	1 (6.7%)	3 (9.1%)	
<0.9 kg/kg/m^2^BS	14 (93.3%)	30 (90.9%)	1.000
Muscle Quality Index (MQI) Adjusted for SMMI			
≥3.3 kg/kg/m^2^BS	5 (33.3%)	8 (24.2%)	
<3.3 kg/kg/m^2^BS	10 (66.7%)	25 (75.8%)	0.509
Muscle Quality Index (MQI) Adjusted for FMI			
≥3 kg/kg/m^2^BS	-	-	
<3 kg/kg/m^2^BS	15 (100%)	33 (100%)	-

* Significant with a *p*-value < 0.05, determined using the chi-square and Fisher’s exact tests, with expected values of less than 5.

**Table 4 biomedicines-12-00280-t004:** Comparison of the levels of adipocytokines and metabolic biomarkers in healthy women and women with breast cancer according to the BMI obesity classification.

	Healthy Women (Control Group, n = 69)				Breast Cancer (Case Group, n = 87)			
Adipocytokines and Metabolic Biomarkers	Normal Weight *n =* 22 Mean (SD)	Overweight *n =* 32 Mean (SD)	Obesity *n =* 15 Mean (SD)	*p*-Value *	Normal Weight *n =* 26 Mean (SD)	Overweight *n =* 28 Mean (SD)	Obesity *n =* 33 Mean (SD)	*p*-Value *
C-peptide (pg/mL) Md (IQI)	189.9 (144–216)	221 (186–264)	215 (167–315)	**0.036**	315.5 (161–389)	358.4 (274–467)	382 (242–633)	0.163
GIP (pg/mL)	15.61 (6.26)	19.34 (25.26)	16.53 (5.99)	0.732	57.2 (31–73)	47 (35–60)	51.3 (40–75)	0.709
GLP-1 (pg/mL)	14.85 (4.36)	14.51 (4.68)	14.30 (3.18)	0.926	63.6 (6.6–142.5)	74.8 (50–131)	53 (0.01–138)	0.361
Ghrelin (pg/mL) Md (IQI)	71.7 (65–107)	65 (43–77)	48.6 (39–70)	**0.008**	149.5 (75–408)	168 (59–251)	157 (55–354)	0.778
Glucagon (pg/mL) Md (IQI)	187.7 (155–198)	192 (177–200)	185 (179–192)	0.508	321.44 (203.2)	335 (184.81)	252.96 (201)	0.310
Insulin (mU/mL)	0.97 (0.49)	1.21 (0.65)	1.43 (0.93)	0.125	17.8 (11–34)	15.6 (9.3–27)	17.3 (11–28.3)	0.641
HOMA-IR (Homeostasis Model Assessment of Insulin Resistance)	0.22 (0.14)	0.29 (0.20)	0.31 (0.22)	0.256	2.48 (5.71)	2.11 (5.83)	3.79 (10.40)	0.678
Leptin (ng/mL) Md (IQI)	1.6 (0.6–3.2)	2.4 (1.3–3.6)	4.1 (2–10.7)	**0.007**	5.68 (7.89)	4.78 (4.34)	12.23 (15.31)	**0.014**
Adiponectin (μg/mL)	4.4 (1.8–8)	2.5 (1.4–4.4)	3.2 (1.6–4.4)	0.378	9.7 (7.3–18.2)	6.1 (3–13.4)	8.6 (2.7–18)	0.227
Resistin (ng/mL) Md (IQI)	3 (1–28)	5 (1–28)	4.5 (1.2–28)	0.521	2.26 (1.62)	2.37 (1.36)	4.14 (8.51)	0.311
Visfatin (ng/mL)	0.85 (0.2)	0.89 (0.17)	0.86 (0.14)	0.732	1.4 (1.1)	1.3 (0.9)	2.06 (4.9)	0.560
PAI-1 (ng/mL) Md (IQI)	292.6 (89–2194)	202 (103–2217)	376 (133–2217)	0.770	20.4 (12.6–255)	19 (13.6–37)	22 (12–295.4)	0.910
Adipsin (μg/mL)	0.42 (0.23)	0.49 (0.21)	0.51 (0.21)	0.313	0.90 (0.83)	0.86 (0.70)	0.91 (0.91)	0.975

Values with a nonparametric distribution are expressed as the median (Md) and interquartile interval (IQI). * Significant with a *p*-value < 0.05, as determined using ANOVA and Bonferroni tests to identify differences between groups.

**Table 5 biomedicines-12-00280-t005:** Comparison of the Biochemical Marker cutoff between healthy women and women with breast cancer, with normal and abnormal values.

**Normal-Weight Women (n = 48)**			
**Biochemical Marker, Normal Values, Cutoff**	**Control Group** ***n =* 22,** ** Mean (SD)**	**Breast Cancer Group** ***n =* 26,** ** Mean (SD)**	** *p* ** **-Value ***
C-peptide			
<216 pg/mL (normal values)	18 (85.7%)	9 (34.6%)	
≥216 pg/mL (high values)	3 (14.3%)	17 (65.4%)	0.001
Raw Odds Ratio (OR) for Cancer Risk in Normal-Weight Women	OR (CI95%); *p*-value		
High C-peptide values	11.1 (2.6–49); 0.001		
GIP (Glucose-Dependent Insulinotropic Polypeptide)			
<18 pg/mL (normal values)	15 (71.4%)	0 (0%)	
≥18 pg/mL (high values)	6 (28.6%)	26 (100%)	0.000
Raw Odds Ratio (OR)	OR (CI95%); *p*-value		
High GIP values	5.3 (2.6–11); 0.000		
GLP-1 (Glucagon-Like Peptide-1)			
<18 pg/mL (normal values)	14 (66.7%)	8 (30.8%)	
≥18 pg/mL (high values)	7 (33.3%)	18 (69.2%)	0.020
Raw Odds Ratio (OR)	OR (CI95%); *p*-value		
High GLP-1 values	4.5 (1.3–15.4); 0.020		
PAI-1 (Plasminogen Activator Inhibitor-1)			
≥94 ng/mL (normal values)	16 (76.2%)	7 (26.9%)	
<94 ng/mL (high values)	5 (23.8%)	19 (73.1%)	0.001
Raw Odds Ratio (OR)	OR (CI95%); *p*-value		
	8.7 (2.3–32.7); 0.001		
**Overweight Women (n = 60)**			
**Age and Anthropometrical Indicators**	**Control Group** ***n =* 32,** ** mean (SD)**	**Breast Cancer Group** ***n =* 28,** ** mean (SD)**	** *p* ** **-Value ***
C-peptide			
<216 pg/mL (normal values)	16 (48.5%)	4 (14.3%)	
≥216 pg/mL (high values)	17 (51.5%)	24 (85.7%)	0.006
Raw Odds Ratio (OR)	OR (CI95%); *p*-value		
High C-peptide values	5.6 (1.6–20); 0.006		
GIP (Glucose-Dependent Insulinotropic Polypeptide)			
<18 pg/mL (normal values)	24 (72.7%)	0 (0%)	
≥18 pg/mL (high values)	9 (27.3%)	28 (100%)	0.000
Raw Odds Ratio (OR)	OR (CI95%); *p*-value		
High GIP values	4.1 (2.3–7.3); 0.000		
GLP-1 (Glucagon-Like Peptide-1)			
<18 pg/mL (normal values)	26 (78.8%)	6 (21.4%)	
≥18 pg/mL (high values)	7 (24.1%)	22 (78.6%)	0.000
Raw Odds Ratio (OR)	OR (CI95%); *p*-value		
High GLP-1 values	13.6 (4–46.6); 0.000		
PAI-1 (Plasminogen Activator Inhibitor-1)			
≥94 ng/mL (normal values)	27 (81.8%)	5 (17.9%)	
<94 ng/mL (high values)	6 (18.2%)	23 (82.1%)	0.000
Raw Odds Ratio (OR)	OR (CI95%); *p*-value		
	20.7 (5.6–76.8); 0.000		
**Obese Women (n = 48)**			
**Age and Anthropometrical Indicators**	**Control Group** ***n =* 15,** ** mean (SD)**	**Breast Cancer Group** ***n =* 33,** ** mean (SD)**	** *p* ** **-Value ***
C-peptide			
<216 pg/mL (normal values)	8 (53.3%)	4 (12.1%)	
≥216 pg/mL (high values)	7 (46.7%)	29 (87.9%)	0.004
Raw Odds Ratio (OR)	OR (CI95%); *p*-value		
High C-peptide values	8.3 (1.9–35.5); 0.004		
GIP (Glucose-Dependent Insulinotropic Polypeptide)			
<18 pg/mL (normal values)	9 (60%)	4 (12.1%)	
≥18 pg/mL (high values)	6 (40%)	29 (87.9%)	0.001
Raw Odds Ratio (OR)	OR (CI95%); *p*-value		
	10.9 (2.5–47.3); 0.001		
GLP-1 (Glucagon-Like Peptide-1)			
<18 pg/mL (normal values)	12 (85.7%)	15 (45.5%)	
≥18 pg/mL (high values)	2 (14.3%)	18 (54.5%)	0.022
Raw Odds Ratio (OR)	OR (CI95%); *p*-value		
	7.2 (1.4–37.3); 0.022		
PAI-1 (Plasminogen Activator Inhibitor-1)			
≥94 ng/mL (normal values)	12 (80%)	11 (33.3%)	
<94 ng/mL (high values)	3 (20%)	22 (66.7%)	0.004
Raw Odds Ratio (OR)	OR (CI95%); *p*-value		
	8 (1.9–34.4); 0.004		

* Significant with a *p*-value < 0.05, as determined using chi-square and Fisher’s exact tests, with expected values of less than 5.

**Table 6 biomedicines-12-00280-t006:** Comparison of the anthropometric parameters between healthy women and patients with breast cancer.

**Normal-Weight Women (n = 48)**			
**Age and Anthropometrical Indicators**	**Control Group** ***n =* 22,** ** Mean (SD)**	**Breast Cancer Group** ***n =* 26,** ** Mean (SD)**	** *p* ** **-Value ***
Age (years)	48.91 (12.37)	51.27 (11.82)	0.503
Body Mass Index (BMI) (kg/m^2^)	23.20 (1.94)	22.35 (1.57)	0.097
Appendicular Skeletal Muscle Index (ASMI) (kg/m^2^)	6.46 (0.46)	6.22 (0.55)	0.108
Fat Mass Index (FMI) (kg/m^2^)	7.18 (2.25)	6.91 (1.25)	0.620
Fat Mass Percentage (%)	30.63 (8.43)	30.78 (4.21)	0.940
Max Handgrip Strength (kg)	25.49 (5.48)	21.57 (4.12)	**0.007**
Muscle Quality Index (HGS/BMI = MQIM) (kg/BMI)	1.11 (0.25)	0.97 (0.21)	0.056
Muscle Quality Index (HGS/SMMI = MQI per ASMMI) (kg/SMMI)	3.95 (0.86)	3.51 (0.83)	0.079
Muscle Quality Index According to Fat Mass Index (kg/FMI)	2.66 (0.81)	3.35 (3.68)	0.392
**Overweight Women (n = 60)**			
**Age and Anthropometrical Indicators**	**Control Group** ***n =* 32,** ** mean (SD)**	**Breast Cancer Group ** ***n =* 28,** ** mean (SD)**	** *p* ** **-Value ***
Age (years)	51.50 (6.65)	54.14 (9.85)	0.223
Body Mass Index (BMI) (kg/m^2^)	26.88 (1.09)	27.99 (1.23)	0.000
Appendicular Skeletal Muscle Index (ASMI) (kg/m^2^)	6.92 (0.43)	7.05 (0.75)	0.436
Fat Mass Index (FMI) (kg/m^2^)	9.93 (1.42)	10.55 (1.17)	0.072
Fat Mass Percentage (%)	36.91 (4.87)	37.64 (3.32)	0.505
Max Handgrip Strength (kg)	26.50 (4.82)	22.52 (6.50)	0.009
Muscle Quality Index (HGS/BMI = MQIM) (kg/BMI)	0.99 (0.17)	0.81 (0.23)	0.001
Muscle Quality Index (HGS/SMMI = MQI per ASMMI) (kg/SMMI)	3.83 (0.65)	3.22 (0.97)	0.005
Muscle Quality Index by Fat Mass Index (kg/FMI)	2.87 (1.09)	2.16 (0.95)	0.010
**Obese Women (n = 48)**			
**Age and Anthropometrical Indicators**	**Control Group** ***n =* 15,** ** mean (SD)**	**Breast Cancer Group** ***n =* 33,** ** mean (SD)**	** *p* ** **-Value ***
Age (years)	51.60 (9.25)	54.64 (11.71)	0.381
Body Mass Index (BMI) (kg/m^2^)	34.59 (4.17)	34.35 (4.28)	0.859
Appendicular Skeletal Muscle Index (ASMI) (kg/m^2^)	7.90 (0.94)	7.98 (0.94)	0.780
Fat Mass Index (FMI) (kg/m^2^)	15.49 (3.25)	15.20 (2.95)	0.762
Fat Mass Percentage (%)	44.43 (4.80)	43.98 (3.71)	0.750
Max Handgrip Strength (kg)	24.04 (4.05)	22.01 (5.55)	0.211
Muscle Quality Index (HGS/BMI = MQIM) (kg/BMI)	0.70 (0.14)	0.65 (0.18)	0.321
Muscle Quality Index (HGS/SMMI = MQI per ASMMI) (kg/SMMI)	3.08 (0.59)	2.78 (0.74)	0.176
Muscle Quality Index by Fat Mass Index (kg/FMI)	1.85 (0.93)	2.34 (0.98)	0.111

* Significant with a *p*-value < 0.05, as determined using chi-square and Fisher’s exact tests, with expected values of less than 5.

**Table 7 biomedicines-12-00280-t007:** Comparison of adipocytokines and metabolic biomarker levels between healthy women and patients with breast cancer according to BMI obesity classification.

**Normal-Weight Women (n = 48)**			
**Biochemical Indicators and Adipocytokines**	**Control Group** ***n =* 22,** ** Mean (SD)**	**Breast Cancer Group ** ***n =* 26,** ** Mean (SD)**	** *p* ** **-Value ***
C-peptide (pg/mL)	189.92 (143.89–215.84)	315.47 (161.55–388.77)	0.015
GIP (pg/mL)	15.62 (10.76–18.26)	57.18 (30.93–73.05)	0.000
GLP-1 (pg/mL)	14.58 (11.31–17.88)	63.60 (6.61–142.50)	0.018
Ghrelin (pg/mL)	71.71 (65.01–107.11)	149.47 (74.96–408.27)	0.005
Glucagon (pg/mL)	187.68 (154.97–197.74)	313.69 (82.27–541.49)	0.047
Insulin (mU/mL)	1.03 (0.64–1.18)	2.96 (1.82–5.63)	0.000
HOMA-IR (Homeostasis Model Assessment of Insulin Resistance)	0.22 (0.13–0.27)	0.61 (0.38–2.20)	0.000
Leptin (ng/mL)	1.57 (0.63–3.18)	3.04 (1.56–6.68)	0.005
Adiponectin (μg/mL)	4.37 (1.78–8.11)	9.75 (7.31–18.20)	0.002
Resistin (ng/mL)	2.89 (0.98–28.20)	1.69 (1.05–2.80)	0.264
Visfatin (ng/mL)	857.99 (769.34–893.27)	852.23 (646.10–1992.63)	0.918
PAI-1 (ng/mL)	292.64 (88.98–2193.84)	20. 38 (12.6–255.44)	0.000
Adipsin (μg/mL)	0.36 (0.26–0.48)	0.90 (0.00–1.29)	0.051
**Overweight Women (n = 60)**			
**Biochemical Indicators and Adipocytokines**	**Control Group** **n = 32,** ** mean (SD)**	**Breast Cancer Group** **n = 28,** ** mean (SD)**	** *p* ** **-Value ***
C-peptide (pg/mL)	220.63 (186.53–264.47)	358.44 (273.77–467.28)	0.000
GIP (pg/mL)	14.19 (10.67–18.77)	47.10 (34.81–60.37)	0.000
GLP-1 (pg/mL)	14.79 (12.73–17.34)	74.78 (50.35–130.68)	0.000
Ghrelin (pg/mL)	64.97 (43.15–77.06)	168.33 (58.82–250.74)	0.000
Glucagon (pg/mL)	191.81 (177.49–200.26)	317.44 (247.25–522.92)	0.000
Insulin (mU/mL)	1.18 (0.71–1.63)	2.60 (1.56–4.50)	0.000
HOMA-IR (Homeostasis Model Assessment of Insulin Resistance)	0.26 (0.15–0.38)	0.62 (0.35–1.07)	0.000
Leptin (ng/mL)	2.41 (1.26–3.62)	3.23 (2.09–5.36)	0.031
Adiponectin (μg/mL)	2.49 (1.45–4.37)	6.12 (2.90–13.37)	0.004
Resistin (ng/mL)	4.80 (1.23–28.20)	2.15 (1.42–3.38)	0.044
Visfatin (ng/mL)	875.26 (812.76–983.61)	786.93 (656.20–1992.90)	0.790
PAI-1 (ng/mL)	201 (103–221.73)	18. 74 (13. 62–36.83)	0.000
Adipsin (μg/mL)	0.47 (0.37–0.59)	0.84 (0.19–1.29)	0.059
**Obese Women (n = 48)**			
**Biochemical Indicators and Adipocytokines**	**Control Group** **n = 15,** ** mean (SD)**	**Breast Cancer Group** **n = 33,** ** mean (SD)**	** *p* ** **-Value ***
C-peptide (pg/mL)	215.84 (166.71–314.98)	382.03 (241.57–632.62)	0.003
GIP (pg/mL)	17.53 (12.08–18.67)	51.35 (39.63–74.47)	0.000
GLP-1 (pg/mL)	14.00 (11.87–14.93)	52.58 (0.01–138.46)	0.623
Ghrelin (pg/mL)	48.61 (38.97–70.13)	156.85 (55.23–353.85)	0.003
Glucagon (pg/mL)	185.34 (179.09–192.21)	302.32 (70.70–406.93)	0.617
Insulin (mU/mL)	1.18 (0.78–1.61)	2.89 (1.75–4.72)	0.001
HOMA-IR (Homeostasis Model Assessment of Insulin Resistance)	0.26 (0.19–0.35)	0.66 (0.40–0.99)	0.001
Leptin (ng/mL)	4.09 (1.98–10.68)	6.05 (2.68–14.76)	0.238
Adiponectin (μg/mL)	3.17 (1.64–4.37)	8.61 (2.67–18.07)	0.004
Resistin (ng/mL)	4.54 (1.96–24.20)	2.25 (1.61–3.92)	0.034
Visfatin (ng/mL)	860.53 (725.09–993.53)	847.40 (623.55–2109.35)	0.632
PAI-1 (ng/mL)	376.32 (132.8–2217.31)	21.89 (12.35–295.38)	0.001
Adipsin (μg/mL)	0.47 (0.45–0.52)	0.92 (0.03–1.39)	0.469

* Significant with a *p*-value < 0.05, as determined using the *t*-test for independent samples to identify differences between groups for variables with parametric distribution. In groups with a nonparametric distribution, the data are described as the median and interquartile interval (IQI), and the *p*-value was estimated using the Mann–Whitney U test.

**Table 8 biomedicines-12-00280-t008:** Multivariate analysis of the risk of breast cancer.

**Total Population**	
**Biochemical Risk Factors—Adjusted Odds Ratio (OR)**	** * OR (CI95%); p-value * **
**Elevated GIP**	36.5 (11.4–115.5); **0.000**
**Elevated GLP-1**	4.16 (1.7–10.4); **0.002**
**Obese Women (n = 48)**	
**Biochemical Risk Factors—Adjusted Odds Ratio (OR)**	** * OR (CI95%); p-value * **
**Elevated GIP**	8.8 (1.8–44); **0.008**
**Elevated GLP-1**	6.5 (1.1–38.7); **0.041**

* Logistic regression: C-peptide, glucagon, and PAI-1 had roles as confounding variables with respect to the intestinal incretins (GIP and GLP-1); significant with a *p*-value < 0.05 and CI95%, both with limits above or below the unit (1).

## Data Availability

The datasets generated and/or analyzed during the current study are not publicly available because they are the property of the Instituto Mexicano del Seguro Social. Institutional and federal regulations restrict unlimited access to personal data, but they are available from the corresponding author upon reasonable request with prior authorization from the institution.

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
