# Peer review of "Role of Incretins in Muscle Functionality, Metabolism, and Body Composition in Breast Cancer: A Metabolic Approach to Understanding This Pathology"

_biomedicines, 2024, doi:10.3390/biomedicines12020280_

Round 1
Reviewer 1 Report (Previous Reviewer 1)
Comments and Suggestions for Authors
Abstract
In this section the future reader can understand what experiment has been carried out. There is enough information about who participated, how they did it, for how long and under which treatments. The main results and the main conclusion of this work are also presented.
Introduction
This section reviews some research similar to that carried out in the experiment in this article, specifically those that present certain similarities and can be compared with the final results reached in this one.
Lines 24-26. Why is it in blue...? Briefly report on some of the main or most widely used evaluation tools referred to.
Lines 40-43. Ok good definition of the objective
2. Materials and Methods
Ok, good explanation but you need to include the Handgrip force measurement material which then appears with results in some later table.
Body Composition Analysis by BIA, anthropometric measurements, muscular strength, body indexes, and muscle quality calculations adjusted by body surface.
Ok, good explanation of anthropometric assessment.
Blood sample and metabolic markers and adipocytokines levels.
Ok, good explanation on the collection of blood sample and metabolic markers and adipocytokines levels.
2.3. Statistical Analysis
Ok, good explanation
3. Results
Please report on the normality of the distribution of the individual samples as finally in table 2 subgroups of only 15 subjects appear as in Healthy women (Control-Group) obesity.
Please include the effect size in table 3.
There are tables and text in yellow and grey that I don't know what they mean. This should be corrected.
4. Discussion
This is undoubtedly the best section of the whole article. The authors make a real effort to gather the previous information regarding the object of study and manage to compare it with the results that have seen the light of day in this article. In addition, at the end, albeit in a very generic way, they make some practical recommendations resulting from the findings of this research.
Author Response
January 11, 2024
Prof. Dr. Felipe Fregni
Editor-in-Chief
Biomedicines
- Neuromodulation Center and Center for Clinical Research Learning, Spaulding Rehabilitation Hospital and Massachusetts General Hospital, Harvard Medical School, Boston, MA 02114, USA
- Department of Epidemiology, Harvard T.H. Chan School of Public Health, Boston, MA 02115, USA
Dr. Sandip Mukherjee
Guest Editor
Special Issue "Recent Advances in Obesity-Related Metabolic Diseases"
Division of Geriatrics and Nutritional Science, Washington University in St. Louis, 660 S. Euclid Ave., St. Louis, MO 63110, USA
PRESENT
Please find enclosed the corrected version of our manuscript titled " Role of Incretins on Muscle Functionality, Metabolism and Body Composition in Breast Cancer: A Metabolic View Understanding This Pathology ", which we re-submit you to consider for publication as an original research article in the Special Issue "Recent Advances in Obesity-Related Metabolic Diseases" in the Journal “Biomedicines.” This is the modified version of our paper on Metabolic characteristics concerning Breast Cancer, comparing anthropometric and biochemical parameters between healthy and breast cancer women, according to the reviewers’ comments and suggestions.
The results in our population emphasize the relationship of metabolic and biochemical markers with skeletal muscle mass and its functional characteristics in breast cancer.
We present the comments of each reviewer, coupled with the response to every comment, and highlight the comments in a different color per each reviewer (per reviewer one, the changes are highlighted in blue; per reviewer 2 in green, and per reviewer 3 in yellow) in the body of the manuscript:
Journal: Biomedicines (ISSN 2227-9059)
Manuscript ID: biomedicines-2718998
Type: Article
Title: Role of Incretins on Muscle Functionality, Metabolism and Body Composition in Breast Cancer: A Metabolic View Understanding This Pathology
Principio del formulario
REVIEWER 1
Open Review
(x) I would not like to sign my review report
( ) I would like to sign my review report
Quality of English Language
(x) I am not qualified to assess the quality of English in this paper
( ) English very difficult to understand/incomprehensible
( ) Extensive editing of English language required
( ) Moderate editing of English language required
( ) Minor editing of English language required
( ) English language fine. No issues detected
|
Yes |
Can be improved |
Must be improved |
Not applicable |
|
|
Does the introduction provide sufficient background and include all relevant references? |
(x) |
( ) |
( ) |
( ) |
|
Are all the cited references relevant to the research? |
(x) |
( ) |
( ) |
( ) |
|
Is the research design appropriate? |
( ) |
(x) |
( ) |
( ) |
|
Are the methods adequately described? |
( ) |
(x) |
( ) |
( ) |
|
Are the results clearly presented? |
( ) |
(x) |
( ) |
( ) |
|
Are the conclusions supported by the results? |
(x) |
( ) |
( ) |
( ) |
Comments and Suggestions for Authors
Abstract
In this section the future reader can understand what experiment has been carried out. There is enough information about who participated, how they did it, for how long and under which treatments. The main results and the main conclusion of this work are also presented.
Answer:
Thanks for the comments about the abstract; we left it without changes.
Introduction
This section reviews some research similar to that carried out in the experiment in this article, specifically those that present certain similarities and can be compared with the final results reached in this one.
Lines 24-26. Why is it in blue...? Briefly report on some of the main or most widely used evaluation tools referred to.
Lines 40-43. Ok good definition of the objective
Answer:
Lines 24-26 were in blue because they contain the modifications requested by the first of the reviewers. We highlighted the comments per each reviewer in a different color. We erased the highlight color for this specific phrase.
Thanks for your general comments about the introduction section.
- Materials and Methods
Ok, good explanation but you need to include the Handgrip force measurement material which then appears with results in some later table.
Answer:
The beginning of the phrases that describe hand grip strength between lines 20 to 25, we described the Dynamometer used to measure handgrip strength, and the complete procedures to get this variable. We reproduce the phrases below:
“We got the muscular strength through a Jamar Plus+ Digital hand dynamometer (Patterson Medical Supply, Cedarburg, WI, USA), according to the recommendations of the American Association of Hand Therapists. The patients held the device and compressed it with maximum force to obtain a maximum contraction. The test was repeated three times for each hand, with one-minute rest intervals between measurements. The highest result of all tests was recorded”.
Body Composition Analysis by BIA, anthropometric measurements, muscular strength, body indexes, and muscle quality calculations adjusted by body surface.
Ok, good explanation of anthropometric assessment.
Blood sample and metabolic markers and adipocytokines levels.
Ok, good explanation on the collection of blood sample and metabolic markers and adipocytokines levels.
Answer:
Thanks for your comments in the four previous paragraphs.
2.3. Statistical Analysis
Ok, good explanation
Answer:
Thanks for your comments concerning the statistical analysis.
- Results
Please report on the normality of the distribution of the individual samples as finally in table 2 subgroups of only 15 subjects appear as in Healthy women (Control-Group) obesity.
Please include the effect size in table 3.
Answer:
The normality test for subgroups of women without cancer divided by nutritional status had a different pattern for each of the studied variables.
We decided to use the Hedges g to calculate the sample size effect because the formula includes the sample size and the mean and variance, and we declare the value for Hedges g; additionally, in the variables with non-parametric distribution, we use the median and interquartile intervals to minimize the effect of sample size and outliers; to calculate the p-value, we perform non-parametric alternatives in the inferential tests. The sample size that we get in obese women without other pathologic conditions grouped as health obesity is a rare group difficult to recruit study subjects, and this is one of the limitations of our study group. The values of the Hedges G, Cohen or Eta and epsilon-squared in the numerical variables with a parametrical distribution where we identified statistical differences through mean and standard deviation was between 0.23 and 0.5 (page 5, lines 12 to 15). The rest of the numerical variables with non-parametrical distribution were evaluated using median and interquartile intervals, so we could not perform any formula for calculating the sample size effect. The values for Hedges G and Cohen test are included in an additional Excel file.
There are tables and text in yellow and grey that I don't know what they mean. This should be corrected.
Answer:
Lines that are highlighted are because they contain the modifications requested by the reviewers. We highlighted the comments per each reviewer in a different color. We erased the highlight color for each specific phrase but highlighted the changes requested by each reviewer. For example, the changes you requested regarding results are highlighted in blue.
- Discussion
This is undoubtedly the best section of the whole article. The authors make a real effort to gather the previous information regarding the object of study and manage to compare it with the results that have seen the light of day in this article. In addition, at the end, albeit in a very generic way, they make some practical recommendations resulting from the findings of this research.
Answer:
Thanks for your comments concerning the Discussion section.
Submission Date
28 December 2023
Date of this review
09 Jan 2024 10:35
Reviewer 2 Report (Previous Reviewer 3)
Comments and Suggestions for Authors
I commend the authors on the completion of this manuscript. It is on an important topic. I have some minor concerns highlighted below.
Specific Comments
Methods
Pag 4. Line 3. Please, explain the acronym PAI-1. It is the first time it appears in the text here.
Pag 4. Line 32. “The blood sample was extracted from patients with 9 to 12 hours of fasting, and the study subjects can drink water and peripheral venous blood from basilic or cephalic arm veins.” Please, syntax review.
Results
Pag 9. Line 6. This in the Healthy women group. You have to specify.
Table 5. Please include the OR (CI95%) only in the corresponding column, I suppose Cancer women.
Pag. 11. Line 16. Please, remove table 6, from this text. You have not presented, the intra-group post-hoc analysis p values in the table 6.
Lines 5-9. Please remove. “Women who were overweight showed differences in HGS and MQI adjusted by BMI, 5 Appendicular Skeletal Muscle Mass Index (ASMMI), and Fat Mass Index (FMI) between 6 those with and without cancer. However, in the group of obese women, no significant 7 differences were found between those with breast cancer and those without (refer to Table 8 6).” At this point you are mixing the intragroup with the intergroup analysis.
Table 8. Multivariate analysis of risk of breast cancer. Please include the OR (CI95%) only in the corresponding column, I suppose Cancer women.
Comments on the Quality of English Language
Minor editing of English language required
Author Response
Journal: Biomedicines (ISSN 2227-9059)
Manuscript ID: biomedicines-2718998
Type: Article
Title: Role of Incretins on Muscle Functionality, Metabolism and Body Composition in Breast Cancer: A Metabolic View Understanding This Pathology
Principio del formulario
REVIEWER 2
Open Review
(x) I would not like to sign my review report
( ) I would like to sign my review report
Quality of English Language
( ) I am not qualified to assess the quality of English in this paper
( ) English very difficult to understand/incomprehensible
( ) Extensive editing of English language required
( ) Moderate editing of English language required
(x) Minor editing of English language required
( ) English language fine. No issues detected
|
Yes |
Can be improved |
Must be improved |
Not applicable |
|
|
Does the introduction provide sufficient background and include all relevant references? |
(x) |
( ) |
( ) |
( ) |
|
Are all the cited references relevant to the research? |
(x) |
( ) |
( ) |
( ) |
|
Is the research design appropriate? |
(x) |
( ) |
( ) |
( ) |
|
Are the methods adequately described? |
( ) |
( ) |
(x) |
( ) |
|
Are the results clearly presented? |
( ) |
( ) |
(x) |
( ) |
|
Are the conclusions supported by the results? |
(x) |
( ) |
( ) |
( ) |
Comments and Suggestions for Authors
I commend the authors on the completion of this manuscript. It is on an important topic. I have some minor concerns highlighted below.
Specific Comments
Methods
Pag 4. Line 3. Please, explain the acronym PAI-1. It is the first time it appears in the text here.
Answer:
We Add the meaning for the acronym. It is highlighted in green.
Pag 4. Line 32. “The blood sample was extracted from patients with 9 to 12 hours of fasting, and the study subjects can drink water and peripheral venous blood from basilic or cephalic arm veins.” Please, syntax review.
Answer:
We enhance the syntaxis and redaction: The blood sample was extracted from patients with 9 to 12 hours of fasting. The fasting did not imply avoid that study subjects can drink water. the peripheral venous blood was extracted from basilic or cephalic arm veins.
Results
Pag 9. Line 6. This in the Healthy women group. You have to specify.
Answer:
We specified that it is in healthy women. Thanks for identify it. This is the phrase (page 9; lines 5 and 6):
“Lower levels of ghrelin were observed in non-comorbidities women with overweight and obesity compared with healthy normal-weight women (71.7% versus 65% and 48.6%, respectively)”.
Table 5. Please include the OR (CI95%) only in the corresponding column, I suppose Cancer women.
Answer:
We put the OR (CI 95%) and p-value in the center of each row (highlighted in green) because the odds represent risk estimations of cancer development of cancer in women without cancer but with the abnormal rise in a biochemical marker proposed such as risk factor.
The odds ratio (OR) is used in case-control studies, and they are a measurement of risk proportion between having or not a risk factor; conversely, the relative risk used in cohorts and clinical assays permits getting the real rate of incidence for each risk factor. The OR is an approximation to relative risk.
Pag. 11. Line 16. Please, remove table 6, from this text. You have not presented, the intra-group post-hoc analysis p values in the table 6.
Answer:
We changed the paragraph:
“However, in the group of obese women, no significant differences were found between those with breast cancer and those without (p values £0.05)”.
Lines 5-9. Please remove. “Women who were overweight showed differences in HGS and MQI adjusted by BMI, 5 Appendicular Skeletal Muscle Mass Index (ASMMI), and Fat Mass Index (FMI) between 6 those with and without cancer. However, in the group of obese women, no significant 7 differences were found between those with breast cancer and those without (refer to Table 8 6).” At this point you are mixing the intragroup with the intergroup analysis.
Answer:
We eliminated the paragraph that you commented on above. This is the current form of the paragraph (page 13; lines 3-4):
The study found a significant difference in HGS and MQI adjusted by BMI in normal-weight women with and without cancer when compared to women who are overweight.
Table 8. Multivariate analysis of risk of breast cancer. Please include the OR (CI95%) only in the corresponding column, I suppose Cancer women.
Answer:
We put the OR (CI 95%) and p-value in the center of each row (highlighted in green) because the odds represent risk estimations of cancer development of cancer in women without cancer but with the abnormal rise in a biochemical marker proposed such as risk factor.
The odds ratio (OR) is used in case-control studies, and they are a measurement of risk proportion between having or not a risk factor; conversely, the relative risk used in cohorts and clinical assays permits getting the real rate of incidence for each risk factor. The OR is an approximation to relative risk.
Comments on the Quality of English Language
Minor editing of English language required
Submission Date
28 December 2023
Date of this review
04 Jan 2024 11:40:26
Reviewer 3 Report (New Reviewer)
Comments and Suggestions for Authors
The study “Role of Incretins on Muscle Functionality, Metabolism and 2 Body Composition in Breast Cancer: A Metabolic View Understanding This Pathology” by Brenda-Eugenia Martínez-Herrera et al. analysed incretin levels, measures of insulin sensitivity, muscle strength and adipokines of women with and without breast cancer.
The manuscript has to be corrected by a native speaker. The discussion has to focus on the current findings. Table 1 may be shown as supplement.
The table legends are unclear and have to better described. In table 2 p-values and * are included, this is unclear. “ANOVA test and Bonferroni test to identify differences between groups.” Which groups?
„Gradually, through new target therapies and early diagnosis [4], the epidemiologic pattern is changing to turn breast cancer into a chronic disease with a high rate of functional disabilities, more than a high lethality entity, affecting active women [5].” It is unclear in this sentence what “active women” means.
“In this frame, the confluence of three chronic conditions, obesity, insulin resistance, and breast cancer, profoundly affects the functionality of treatment-naïve breast cancer women.” This is also unclear, why does breast cancer affect the functionality?
“inflammatory serum,“ what is this?
Please include ethical vote number and date of approval.
“Body Mass Index (BMI): lean (<25 kg/m2BS), overweighted 1 (25 - <30kg/m2BS), and obese (30 kg/m2BS).” Please correct
“Based on our previously published data obtained in healthy women over 40 years 6 old, we took the reference values for HOMA IR (0.78), insulin (2.6 mU/mL), leptin (27.5 7 ng/mL), adiponectin (17.68 mg/mL), resistin (0.59 ng/mL), visfatin (1.18 ng/mL), and adipsin (0.91 mg/mL)” please check the units, at least the unit for adiponectin is not correct.
“according to the recommendations of the American Association of Hand Therapists.” Please add a reference
PAI-1 abbreviation has to be defined when used the first time
Which assay was used for adiponectin analysis?
What does “adjusted Bonferroni test” indicate?
“significance < 0.2“ is this correct?
Adiponectin is mentioned at page 6 but not included in table 1
“nutritional state was not influenced by the age” how was this determined?
Identical data are used for e.g. table 5 and 7. This is confusing and not appropriate. Please reduce the number of tables.
Though obesity is a risk factor for breast cancer there was no difference in the current cohort. Please explain.
The authors can only describe the differences between patients with breast cancer and controls. This is not a prospective studies and risk factors can not be identified.
“Incretins, such as GIP and GLP-1, are essential in regulating insulin secretion in response to nutrient intake.“ However, here patients were in the fasted state and there is no information on the postprandial levels.
Comments on the Quality of English Languagehas to be corrected by a native speaker
Author Response
Journal: Biomedicines (ISSN 2227-9059)
Manuscript ID: biomedicines-2718998
Type: Article
Title: Role of Incretins on Muscle Functionality, Metabolism and Body Composition in Breast Cancer: A Metabolic View Understanding This Pathology
Principio del formulario
REVIEWER 3
Open Review
(x) I would not like to sign my review report
( ) I would like to sign my review report
Quality of English Language
( ) I am not qualified to assess the quality of English in this paper
( ) English very difficult to understand/incomprehensible
(x) Extensive editing of English language required
( ) Moderate editing of English language required
( ) Minor editing of English language required
( ) English language fine. No issues detected
|
Yes |
Can be improved |
Must be improved |
Not applicable |
|
|
Does the introduction provide sufficient background and include all relevant references? |
( ) |
( ) |
(x) |
( ) |
|
Are all the cited references relevant to the research? |
( ) |
( ) |
(x) |
( ) |
|
Is the research design appropriate? |
( ) |
( ) |
(x) |
( ) |
|
Are the methods adequately described? |
( ) |
( ) |
(x) |
( ) |
|
Are the results clearly presented? |
( ) |
( ) |
(x) |
( ) |
|
Are the conclusions supported by the results? |
( ) |
( ) |
(x) |
( ) |
Comments and Suggestions for Authors
The study “Role of Incretins on Muscle Functionality, Metabolism and 2 Body Composition in Breast Cancer: A Metabolic View Understanding This Pathology” by Brenda-Eugenia Martínez-Herrera et al. analysed incretin levels, measures of insulin sensitivity, muscle strength and adipokines of women with and without breast cancer.
The manuscript has to be corrected by a native speaker. The discussion has to focus on the current findings. Table 1 may be shown as supplement.
The table legends are unclear and have to better described. In table 2 p-values and * are included, this is unclear. “ANOVA test and Bonferroni test to identify differences between groups.” Which groups?
Answer:
Between lean, overweight and obese women. We analyzed separately healthy and breast cancer women. We modified the redaction, additionally we sent the manuscript to a native language correction service from MDPI.
„Gradually, through new target therapies and early diagnosis [4], the epidemiologic pattern is changing to turn breast cancer into a chronic disease with a high rate of functional disabilities, more than a high lethality entity, affecting active women [5].” It is unclear in this sentence what “active women” means.
Answer:
Women with social interaction and some of them active workers, nobody isolated or with physical disability. We modified the redaction, additionally we sent the manuscript to a native language correction service from MDPI.
“In this frame, the confluence of three chronic conditions, obesity, insulin resistance, and breast cancer, profoundly affects the functionality of treatment-naïve breast cancer women.” This is also unclear, why does breast cancer affect the functionality?
Answer:
Functionality is a construct used in the evaluation of health-related quality of life through validated scales that measure an ability to function, usually include: physical functioning, role functioning, social functioning, emotional and cognitive functioning. You can see an example in:
King MT, Bell ML, Costa D, Butow P, Oh B. The Quality of Life Questionnaire Core 30 (QLQ-C30) and Functional Assessment of Cancer-General (FACT-G) differ in responsiveness, relative efficiency, and therefore required sample size. J Clin Epidemiol. 2014 Jan;67(1):100-7. doi: https://doi.org/10.1016/j.jclinepi.2013.02.019. Epub 2013 Oct 11. PMID: 24125895.
Available in: https://pubmed.ncbi.nlm.nih.gov/24125895/
“inflammatory serum,“ what is this?
Answer:
We corrected the phrase on page 3, line 5: inflammatory serum markers, and highlighted it in yellow.
Please include ethical vote number and date of approval.
Answer:
The votes of the Institutional Review Board are confidential. The Committee provides the acceptance or approved letter to the researchers. I have included the image:
“Body Mass Index (BMI): lean (<25 kg/m2BS), overweighted 1 (25 - <30kg/m2BS), and obese (30 kg/m2BS).” Please correct
Answer:
“Based on our previously published data obtained in healthy women over 40 years 6 old, we took the reference values for HOMA IR (0.78), insulin (2.6 mU/mL), leptin (27.5 7 ng/mL), adiponectin (17.68 mg/mL), resistin (0.59 ng/mL), visfatin (1.18 ng/mL), and adipsin (0.91 mg/mL)” please check the units, at least the unit for adiponectin is not correct.
Answer:
We realized the conversion of the units. In the Bioplex panel the units are picograms/mL, and adjusted data for adiponectin and other cytokines are reported in:
Sat-Muñoz D, Martínez-Herrera BE, Quiroga-Morales LA, Trujillo-Hernández B, González-Rodríguez JA, Gutiérrez-Rodríguez LX, Leal-Cortés CA, Portilla-de-Buen E, Rubio-Jurado B, Salazar-Páramo M, Gómez-Sánchez E, Delgadillo-Cristerna R, Carrillo-Nuñez GG, Nava-Zavala AH, Balderas-Peña LM. Adipocytokines and Insulin Resistance: Their Role as Benign Breast Disease and Breast Cancer Risk Factors in a High-Prevalence Overweight-Obesity Group of Women over 40 Years Old. Int J Environ Res Public Health. 2022 May 17;19(10):6093. doi: https://doi.org/10.3390/ijerph19106093. PMID: 35627631; PMCID: PMC9140417.
“according to the recommendations of the American Association of Hand Therapists.” Please add a reference
Answer:
We added the reference:
Lopes J, Grams ST, da Silva EF, de Medeiros LA, de Brito CMM, Yamaguti WP. Reference equations for handgrip strength: Normative values in young adult and middle-aged subjects. Clin Nutr. 2018 Jun;37(3):914-918. doi: 10.1016/j.clnu.2017.03.018. Epub 2017 Mar 24. PMID: 28389120.
PAI-1 abbreviation has to be defined when used the first time
Answer:
We Add the meaning for the acronym. It is highlighted in green: (Plasminogen activator inhibitor 1).
Which assay was used for adiponectin analysis?
Answer:
Bioplex:BIO-RAD diabetes panel I and II for human serum (multi-test panel Bio-Plex Pro No. 171A7001M and Bio-Plex Pro 171A7003M, BIO-RAD®, Hercules, CA, USA).
Bio-Plex Pro Human Diabetes Adipsin and Adiponectin Assays #171A7002M. Technical specifications in:
https://www.bio-rad.com/es-mx/sku/171A7002M-bio-plex-pro-human-diabetes-adipsin-adiponectin-assays?ID=171A7002M
What does “adjusted Bonferroni test” indicate?
Answer:
In an intragroup analysis with ANOVA and the Bonferroni complimentary test, the Bonferroni test permits identifying which of the three groups are responsible for the differences when the p-value is <5, and we divided the population in three or more groups
“significance < 0.2“ is this correct?
Answer:
Yes, it is correct to identify variables with p-values for correlation lower than 0.2. It permits identifying all variables with association best probability, and it is part of the procedure to identify which variables must be included in a multivariate analysis. After selecting all variables with a p-value correlation lower than 0.2, we put in a higher to lower correlation (rP or Spearman Rho: 0.9, 0.8, 0.7,…, etcetera) value to include in a step-by-step method as an intervening variable and exclude variables.
Adiponectin is mentioned at page 6 but not included in table 1
Answer:
Adiponectin’s normal value is mentioned in page 4, line 9, it is not mentioned in any table
“nutritional state was not influenced by the age” how was this determined?
Answer:
Because we did not observe significant differences in the age during intragroup analysis (Data shown in Table 2).
Identical data are used for e.g. table 5 and 7. This is confusing and not appropriate. Please reduce the number of tables.
Answer:
Data are different: In Table 5, we compared, in an intragroup analysis (with ANOVA and Bonferroni complimentary test that permits identifying which of three groups are responsible for the differences when p-value is <5), women without breast cancer inside the group according to their nutritional state, and in Table 7, we compare lean, healthy women versus lean cancer women, overweight healthy women versus overweight cancer women, and obese women without cancer versus obese women with breast cancer. There could be similar data, but the statistical analysis and the interpretation are different.
Other reviewers request the preservation of the tables.
Though obesity is a risk factor for breast cancer there was no difference in the current cohort. Please explain.
Answer:
Because the differences between healthy women without obesity and breast cancer women are directly related to the absence of obesity. The effect could be delimitated with a cohort study to long-term follow-up to observe how the gradual increase in weight associated with age is associated with a parallel increase in risks by including the time factor. Still, the possibility of identifying this effect is lost in retrospective and also cross-sectional studies, so we recognize that it is a limitation of our study.
The authors can only describe the differences between patients with breast cancer and controls. This is not a prospective studies and risk factors can not be identified.
Answer:
The odds represent risk estimations (not risk) of cancer development of cancer in women without cancer but with the abnormal rise in a biochemical marker proposed such as risk factor.
The odds ratio (OR) is used in case-control studies, and they are a measurement of risk proportion between having or not a risk factor; conversely, the relative risk used in cohorts and clinical assays permits getting the real rate of incidence for each risk factor. The OR is an approximation to relative risk.
One of the basis of our decision can be found in:
Ranganathan P, Aggarwal R, Pramesh CS. Common pitfalls in statistical analysis: Odds versus risk. Perspect Clin Res. 2015 Oct-Dec;6(4):222-4. doi: https://doi.org/10.4103/2229-3485.167092. PMID: 26623395; PMCID: PMC4640017.
The paper textually says:
The relative risk (also known as risk ratio [RR]) is the ratio of risk of an event in one group (e.g., exposed group) versus the risk of the event in the other group (e.g., nonexposed group). The odds ratio (OR) is the ratio of odds of an event in one group versus the odds of the event in the other group…
… Calculation of risk requires the use of “people at risk” as the denominator. In retrospective (case-control) studies, where the total number of exposed people is not available, RR cannot be calculated and OR is used as a measure of the strength of association between exposure and outcome. By contrast, in prospective studies (cohort studies), where the number at risk (number exposed) is available, either RR or OR can be calculated.
Multiple logistic regression, a frequently used multivariate technique, calculates adjusted ORs and not RRs”.
“Incretins, such as GIP and GLP-1, are essential in regulating insulin secretion in response to nutrient intake.“ However, here patients were in the fasted state and there is no information on the postprandial levels.
Answer:
The incretins levels were taken in a basal state, and this is a valuable point to emphasize: incretins, despite the fasting state, were raised as part of an altered metabolic condition.
I could be part of other clinical designs based on cohort or clinical assay, the dynamic measurement of incretins in fasting, and one and three hours of glucose oral load to analyze the patterns in the incretin levels.
Comments on the Quality of English Language
has to be corrected by a native speaker
Submission Date: 28 December 2023
Date of this review: 10 Jan 2024 07:57:32
Round 2
Reviewer 3 Report (New Reviewer)
Comments and Suggestions for Authors
Authors have addressed the comments in my reviewe. There are 2 minor corrections:
"However, in the group of obese women, no significant differences were found between those with breast cancer and those without (p-values < 0.05)." > has to be used.
Adiponectin« (g/mL) - why is « included?
Author Response
Journal: Biomedicines (ISSN 2227-9059)
Manuscript ID: biomedicines-2718998
Type: Article
Title: Role of Incretins on Muscle Functionality, Metabolism and Body Composition in Breast Cancer: A Metabolic View Understanding This Pathology
Principio del formulario
REVIEWER 3
Open Review
(x) I would not like to sign my review report
( ) I would like to sign my review report
Quality of English Language
( ) I am not qualified to assess the quality of English in this paper
( ) English very difficult to understand/incomprehensible
() Extensive editing of English language required
( ) Moderate editing of English language required
( ) Minor editing of English language required
(x) English language fine. No issues detected
|
Yes |
Can be improved |
Must be improved |
Not applicable |
|
|
Does the introduction provide sufficient background and include all relevant references? |
(x) |
( ) |
( ) |
( ) |
|
Are all the cited references relevant to the research? |
(x) |
( ) |
( ) |
( ) |
|
Is the research design appropriate? |
(x) |
( ) |
( ) |
( ) |
|
Are the methods adequately described? |
(x) |
( ) |
( ) |
( ) |
|
Are the results clearly presented? |
(x) |
( ) |
( ) |
( ) |
|
Are the conclusions supported by the results? |
(x) |
( ) |
( ) |
( ) |
Comments and Suggestions for Authors
Authors have addressed the comments in my reviewe. There are 2 minor corrections:
"However, in the group of obese women, no significant differences were found between those with breast cancer and those without (p-values < 0.05)." > has to be used.
Answer
We change the paragraph: However, in the group of obese women, no significant differences were found between those with breast cancer and those without (p-values > 0.05).
Adiponectin« (g/mL) - why is « included?
Answer
We erase the symbol in all Table 7, in the places where appears the word adiponectin and highlighted in yellow the places where we made the changes. It was a typo mistake.
Submission Date: 28 December 2023
Date of this review: 16 Jan 2024 09:16:27

This manuscript is a resubmission of an earlier submission. The following is a list of the peer review reports and author responses from that submission.
Round 1
Reviewer 1 Report
Comments and Suggestions for Authors
Abstract
Please report the participating subjects and their main characteristics in this section.
1. Introduction
Lines 21-23. In the text you state that several authors have proposed the use of the handgrip as a strength assessment tool ("Some authors have proposed its use..."), however you only include one reference, number 12. You should include more because the existing literature provides a lot of evidence and examples of the use of this handgrip assessment tool.
Lines 24-31, Please consider including this reference: Hidrobo Coello JF. Physical activity for patients diagnosed with cancer. Sports prescription guide for Ecuador. Revista Iberoamericana de Ciencias de la Actividad Física y el Deporte [Internet]. 2020 Dec. 22 [cited 2023 Nov. 20];9(3):18-41. Available from: https://revistas.uma.es/index.php/riccafd/article/view/10100
In general this section describes well the background of the research problem, why it is important to carry out this work and what are the potential effects of the knowledge that this research can provide.
2. Materials and Methods
In general this section describes well what has been done and how it has been done. However, it should include a section describing the characteristics of the subjects.
Body Composition Analysis by BIA, anthropometric measurements, muscular strength, body 5 indexes, and muscle quality calculations adjusted by body surface.
Ok, good description, good work.
2.2. Blood sample and metabolic markers and adipocytokines levels
Please report the conditions under which the subjects underwent the extraction of these biological samples.
2.3. Statistical Analysis
Ok, good description, good job.
3. Results
The information you include between lines 42-52 would look better for future readers if you put them in tables.
3.1. Anthropometric parameters in healthy and breast cancer women group by nutritional state
3.1.1. Age and anthropometric indexes
Ok, good description
3.2. Handgrip strength and muscle quality adjusted by BMI, ASMMI, and FMI
In table 1 remove the abbreviations that you have already included in the text, or put them at the bottom of the page. For example BMI
Please avoid underlining in table 5.
In tables 5, 6, 7 you indicate that there are a total of 168 women (48+60+60), however in the description above and in the tables you indicate that there are 156 (page 5, line 19). This should be explained or corrected or revised.
Please avoid underlining in table 8.
4. Discussion
This is undoubtedly the best section of the paper in which the authors make a real effort to compare the different variables they have assessed with previous research studies. This helps to give solidity to their work and also to give future readers a broad but also deep insight into the new contributions of this work, which means that it can potentially be cited in future papers.
5. Conclusions
The conclusions respond to the classic model of completion of a paper, responding to the extent to which the initial objective has been achieved. However, we must not forget that the aim of science is that it can be applied and therefore my advice is to include some practical recommendations so that doctors, therapists, trainers or the patients themselves can benefit from this new knowledge.
References
Line 22. Please check each and every reference and correct to English the months that are in Spanish.
Reviewer 2 Report
Comments and Suggestions for Authors
My overall opinion on the assessed work entitled: “Role of Incretins on Muscle Functionality, Metabolism and Body Composition in Breast Cancer: A Metabolic View Under-standing This Pathology” is positive.
I consider the manuscript of paper submitted for evaluation in Journal “Biomedicines” to be a highly valuable publication, properly designed as well as carefully and reliably written/edited.
However, I have remarks:
1.The aim of the study does not fully correspond to the conclusions. The conclusions do not constitute a direct response to the research tasks specified in the aim of the study. In addition, the “Conclusions subsection” do not have practical implications.
2.The information contained in the "Introduction subsection" should come from the most modern publications containing up-to-date data. The works cited in the chapter introduction - no. 4., no. 5., no. 15., no. 17., no.18., no. 20. are not the most recent. An "Introduction subsection" should be prepared based on the current, most recent literature data.
3.The authors of the publication did not include inclusion and exclusion criteria in the paper. Mentioned (inclusion and exclusion) criteria must be completed in the publication.
4.Selected and crucial fragments of the text of the “4.Discussion Section” have not been compared/confronted with the research results of other authors i.e. (page 15th - lines 8–16; page 16th - lines 3–4; page 16th - lines 24–26; page 16th - lines 33–38; page 17th - lines 4–9; page 17th - lines 28–32; page 18th - lines 2–4; page 18th - lines 17–20) – it must be supplemented.
5.The most important problem (negative value) of the publication - otherwise excellently written - is the lack of a significantly marked scientific novelty. Authors must in the discussion and in the "conclusion subsection" mark/indicate/emphasize the scientific novelty of the results obtained.
6.Finally, I must add that the preparation of a graphical abstract of the work would significantly increase the value of the manuscript under evaluation. I strongly recommend the graphical abstract to be drawn and added.
Reviewer 3 Report
Comments and Suggestions for Authors
I commend the authors on the completion of this manuscript. It is on an important topic. But I have major concerns highlighted below.
General comments
The results presented are very confusing and very wide in relation with the aim of the study. There are a lot of variables included that have not been presented in the introduction, that are not mentioned in the purpose, that have not been explained in the methods and their analysis in the results is presented confusedly.
Specific Comments
Abstract:
Please explain the acronyms for GIP and GLP-1, the first time they appear in the text of the abstract.
Methods
Line 37. Please, enumerate and explain the method of analysis of the C peptide, GIP, GLP-1, and PAI-1. Please explain the acronym PAI-1
Results
Please, correct the numbers of the subtitles in all the results section.
Lines 42-52. How do you determine the normal values? If they are based on the bibliography, please, include references.
Line 3. 3.1.1. Age and anthropometric indexes. What do you mean with health in this sentence: “without health or nutritional state 3 differences”
You have to rewrite the table 1. This legend is not correct: “*** Significant p-value <0.05 post hoc Bonferroni test between normal weight and overweight and normal weight and obesity groups. † Significant p value <0.05 post hoc Bonferroni test between obesity and overweight, and obesity and normal weight groups. Significant p-value <0.05 post hoc Bonferroni test between normal weight and obesity groups” What is the difference between the *** and the † and the when you speak about “obesity and normal weight groups”. My advice it is to explain all the post-hoc test in the text and add also the p value for these tests.
Line 11: You are not comparing this in the table 1: Women who were overweight showed differences in HGS and MQI adjusted by BMI, Appendicular Skeletal Muscle Mass Index (ASMMI), and Fat Mass Index (FMI) between those with and without cancer. However, in the group of obese women, no significant differences were found between those with breast cancer and those without (refer to Table 14 1).
Table 2. Please correct. “Control Women n = 22 mean (SD) Breast Cancer Women n = 26 mean (SD)” You are not presenting mean(SD)
Line 2 “After establishing cutoffs for normal values in fat mass percentage, fat mass index (FMI), max HGS, and the MQI adjusted by BMI, SMMI, and FMI, we found differences in the max HGS and MQI adjusted by BMI, SMMI and FMI in overweight women in the control and the cancer groups, with lower max-HGS, and lower MQI adjusted by the three above-mentioned anthropometric indexes (see Table 2).” These results here, you are not presenting in table 2.
“Table 3. Comparison of the hematologic parameters between healthy women and patients with 12 breast cancer according to BMI obesity classification.” The legend has the same problem as before. And you are presenting a comparison between BMI classification in each group, not between healthy and patients.
Line 13. Table 4. “between the diferent nutritional states or health conditions” Again you are not comparing health conditions.
Table 5. “Age and Anthropometrical Indicators” Why do you use this in line 2 in table 5?
2.5. Comparison of anthropometric parameters by health state.
Line 3. “The handgrip strength was markedly lower in breast cancer patients [21.57 (±4.12)] compared with control women [25.5 (±5.4)], with a p-value of 0.007. We also noticed a similar trend 5 in the muscle quality index adjusted by BMI, although the difference was not statistically 6 significant (p=0.056).” This in the normal weight group. You have to specify.
Table 7. “Age and Anthropometrical Indicators” Why do you use this in line 2 in table 7?
Table 8. Multivariate analysis of risk of breast cancer.
Which are exactly the variables included in the models (they are different here form the explained in methods section)? And why have you included these variables and not others?
Discussion
Line 15. “However, regarding the progression of the disease, we did not find any significant difference based on clinical stage.” Please, remove, you have not studied this fact.
Avoid the discussion about correlation analysis, because you have not presented these data in the results section.
The discussion is highly unspecific.
Comments on the Quality of English LanguageModerate English review needed.